# Identification of highly oxygenated organic molecules and their role in aerosol formation in the reaction of limonene with nitrate radical

Yindong Guo[1], Hongru Shen[1], Iida Pullinen[2, a], Hao Luo[1,3], Sungah Kang[2], Luc Vereecken[2], Hendrik Fuchs[2], Mattias Hallquist[4], Ismail-Hakki Acir[2,b], Ralf Tillmann[2], Franz Rohrer[2], Jürgen Wildt[2], Astrid Kiendler-Scharr[2], Andreas Wahner[2], Defeng Zhao[1,5,6*], Thomas F. Mentel[2*]

[1]Department of Atmospheric and Oceanic Sciences & Institute of Atmospheric Sciences, Fudan University, 200438, Shanghai, China

[2]Institute of Energy and Climate Research, IEK-8: Troposphere, Forschungszentrum Jülich GmbH, 52425, Jülich, Germany

[3]IRDR ICoE on Risk Interconnectivity and Governance on Weather/Climate Extremes Impact and Public Health, Fudan University, Shanghai 200438, China

[4]Department of Chemistry and Molecular biology, University of Gothenburg, Göteborg, 41258, Sweden

[5]Shanghai Frontiers Science Center of Atmosphere-Ocean Interaction, Fudan University, Shanghai 200438, China

[6]Institute of Eco-Chongming (IEC), 20 Cuiniao Rd., Chongming, Shanghai, 202162, China

[a] Now at: Department of Applied Physics, University of Eastern Finland, Kuopio, 70210, Finland.

[b] Now at: Institute of Nutrition and Food Sciences, University of Bonn, Bonn, 53115, Germany.

*Correspondence to*: Defeng Zhao (dfzhao@fudan.edu.cn), Thomas F. Mentel (t.mentel@fz-juelich.de)

**Abstract.** Nighttime $NO_3$-initiated oxidation of biogenic volatile organic compounds (BVOC) such as monoterpenes is important for the atmospheric formation and growth of secondary organic aerosol (SOA), which has significant impact on climate, air quality and human health. In such SOA formation and growth, highly oxygenated organic molecules (HOM) may be crucial, but their formation pathways and role in aerosol formation have yet to be clarified. Among monoterpenes, limonene is of particular interest for its high emission globally and high SOA yield. In this work, HOM formation in the reaction of limonene with nitrate radical ($NO_3$) was investigated in the SAPHIR chamber (Simulation of Atmospheric PHotochemistry In a large Reaction chamber). About 280 HOM products were identified, grouped into 19 monomer families, 11 dimer families and 3 trimer families. Both closed-shell products and open-shell peroxy radicals ($RO_2\bullet$) were observed, and many of them have not been reported previously. Monomers and dimers accounted for 47 % and 47 % of HOM concentrations, respectively, with trimers making up the remaining 6%. In the most abundant monomer families, $C_{10}H_{15-17}NO_{6-14}$, carbonyl products outnumbered hydroxyl products, indicating the importance of $RO_2\bullet$ termination by unimolecular dissociation. Both $RO_2\bullet$ autoxidation and alkoxy-peroxy pathways were found to be important processes leading to HOM. Time-dependent concentration profiles of monomer products containing nitrogen showed mainly second-generation formation patterns. Dimers were likely formed via the accretion reaction of two monomer $RO_2\bullet$, and HOM-trimers via the accretion reaction between monomer $RO_2\bullet$ and dimer $RO_2\bullet$. Trimers are suggested to play an important role in new

particle formation (NPF) observed in our experiment. A HOM yield of 1.5 %$_{-0.7 \%}^{+1.7 \%}$ was estimated considering only
first-generation products. SOA mass growth could be reasonably explained by HOM condensation on particles
assuming irreversible uptake of ultra-low volatility organic compounds (ULVOC), extremely low volatility organic
compounds (ELVOC) and low volatility organic compounds (LVOC). This work provides evidence for the important
role of HOM formed via the limonene + $NO_3$ reaction in NPF and growth of SOA particles.

## 1 Introduction

The nitrate radical ($NO_3$) is an important nighttime oxidant in tropospheric chemistry, and can reach mixing ratios of
several hundred pptv during nighttime (Seinfeld and Pandis, 2006). It can react with volatile organic compounds
(VOC) and is especially reactive to alkenes, where the nitrate radical can undergo an addition reaction to the C=C
double bond (Finlayson-Pitts and Pitts, 1997; Seinfeld and Pandis, 2006). Biogenic monoterpenes ($C_{10}H_{16}$) are a large
contribution to the alkenes in the atmosphere (Klinger et al., 2002; Guenther et al., 2012), and their major nighttime
loss pathway is reaction with $NO_3$ (Beaver et al., 2012; Rollins et al., 2012; Ayres et al., 2015; Fry et al., 2013). The
chemistry of monoterpenes with $NO_3$ has implications on the cycle of reactive nitrogen and thus on ozone formation
(Brown and Stutz, 2012). Furthermore, since the $NO_3$ radical is formed through the reaction of $NO_2$ with $O_3$, it is
considered to be of anthropogenic origin, and reactions of $NO_3$ with biogenic VOC (BVOC) thus represent an
important interaction between biogenic emissions and anthropogenic emissions.

The reaction of $NO_3$ with monoterpenes can form secondary organic aerosols (SOA), which can have a large

impact on global climate, air quality and human health (Hallquist et al., 2009; Shrivastava et al., 2017). Laboratory
studies showed that monoterpenes have high SOA yields in the reaction with $NO_3$ due to the low volatility of
oxidation products (Ng et al., 2008; Rollins et al., 2009; Fry et al., 2013; Fry et al., 2014; Ayres et al., 2015; Jokinen
et al., 2015; Zhou et al., 2015; Boyd et al., 2015; Nah et al., 2016; Boyd et al., 2017; Slade et al., 2017; Claflin and
Ziemann, 2018; Bates et al., 2022; Dam et al., 2022). Field studies also showed that nighttime $NO_3$-initiated oxidation
of monoterpenes contributes significantly to SOA in forested regions influenced by anthropogenic emissions (Pye et
al., 2010; Rollins et al., 2012; Fry et al., 2013; Ayres et al., 2015; Zhou et al., 2015; Xu et al., 2015; Lee et al., 2016;
Zhang et al., 2018; Chen et al., 2020) and potentially in urban areas due to the extensive usage of so-called volatile
chemical products (VCP) (Nazaroff and Weschler, 2004; Mcdonald et al., 2018). For example, the Southern Oxidant
and Aerosol Study (SOAS) showed that the BVOC + $NO_3$ reactions were a substantial source of SOA (Ayres et al.,
2015; Xu et al., 2015; Lee et al., 2016; Massoli et al., 2018). Therefore, accurate predictions and evaluations of SOA
concentration and thus its climate and environmental effects require a comprehensive understanding of the reactions
of monoterpenes with $NO_3$.

Recently, a class of organic compounds named highly oxygenated molecules (HOM) have been shown to be

critical substances in the SOA formation from BVOC oxidation, particularly monoterpenes, featuring high O/C ratio
and low to extremely low volatility (Ehn et al., 2014; Tröstl et al., 2016; Kirkby et al., 2016; Bianchi et al., 2019).
HOM here refers to compounds formed in the gas phase via autoxidation which contain at least six oxygen atoms
(Bianchi et al., 2019). Most HOM are classified as ULVOC/ELVOC or LVOC (Bianchi et al., 2019) according to the
classification of atmospheric organics based on their volatility (saturation concentration, $C^*$) by Donahue et al. (2012)
(extremely low volatility organic compounds (ELVOC), low volatility organic compounds (LVOC), semi-volatile
organic compounds (SVOC), intermediate volatility organic compounds (IVOC), volatile organic compounds
(VOC)), and a recent update by Schervish and Donahue (2020) (ultra-low volatility organic compounds (ULVOC)).
And thus, HOM can be a substantial contribution to growth of SOA particles through gas-particle partitioning.
A better description of the HOM formation chemistry in the oxidation of monoterpenes by $NO_3$ will improve
our understanding of the role of HOM in SOA formation, as well as the relationship between oxidation products,
SOA formation and reaction systems. Field observation campaigns as well as laboratory experiments have proven
the important contribution of HOM in monoterpene + $NO_3$ SOA (Lee et al., 2016; Faxon et al., 2018). In the SOAS
campaign, HOM-ON (organic nitrates) were identified in both gas and particle phase using a $NO_3^-$-Chemical
Ionization time-of-flight Mass Spectrometer (CI-APi-TOF) and a High Resolution Time-of-Flight Chemical
Ionization Mass Spectrometer (HR-ToF-CIMS) coupled to a Filter Inlet for Gases and AEROsols (FIGAERO).
Species with the sum formula $C_{10}H_{15,17,19}NO_{4-11}$ were observed which are formed through the oxidation of
monoterpenes by $NO_3$ (Lee et al., 2016; Massoli et al., 2018). In a campaign in a boreal forest in Hyytiälä,
measurement using a $NO_3^-$-CI-APi-TOF and positive matrix factor (PMF) analysis showed a nighttime factor of
HOM-ON formed via $NO_3$ oxidation of monoterpenes (Yan et al., 2016). Besides the observations at forested regions,
monoterpene-derived HOM via $NO_3$ oxidation also contribute to organic aerosols in urban regions. For example, Liu
et al. (2021) and Nie et al. (2022) have found that HOM derived from monoterpene nighttime chemistry are important
in megacities in China, especially during summertime. A number of laboratory studies have reported HOM formation
by the oxidation of monoterpenes with $NO_3$. Boyd et al. (2015) observed $C_{10}H_{17}NO_{4/5}$ and $C_{10}H_{15}NO_{5/6}$ in the gas
phase in β-pinene + $NO_3$ experiments using a quadrupole chemical ionization mass spectrometer with $I^-$ as the reagent
ion ($I^-$-CIMS). They proposed possible formation schemes of these ONs. Nah et al. (2016) further detected 5 and 41
HOM-ON in the $NO_3$ oxidation of α-pinene and β-pinene, respectively, such as $C_{10}H_{15/17/19}NO_{4-9}$ in the gas- and
particle-phase using $I^-$-FIGAERO HR-ToF-CIMS. Claflin and Ziemann (2018) provided formation mechanisms for
HOM-ON via gas-phase and particle-phase reactions in the β-pinene + $NO_3$ reaction system, where particle-phase
products were analyzed using reversed-phase high-performance liquid chromatography equipped with a UV−vis
photodiode array detector (HPLC-UV), Electron-Ionization Thermal Desorption Particle Beam Mass Spectrometer
(EI-TDPBMS), Chemical Ionization Finnigan PolarisQ Ion Trap Mass Spectrometer (CI-ITMS), and Electrospray-
Ionization Mass Spectrometer (ESI-MS). Recently, Shen et al. (2021) found a large number of HOM (>150 species)
in the β-pinene + $NO_3$ reaction using $NO_3^-$-CI-APi-TOF. HOM formed in the reaction of four monoterpenes (α-
pinene, β-pinene, Δ-3-carene, and α-thujene) with $NO_3$ were also detected using $NO_3^-$-CI-APi-TOF by Dam et al.
(2022). Bell et al. (2021) found that dimer dinitrates ($C_{20}H_{32}N_2O_{8-13}$) contribute a large portion to SOA from α-pinene
+ $NO_3$ and also detected monomer ON such as $C_{10}H_{15}NO_{5-10}$ and $C_{10}H_{14,16}N_2O_{7-11}$) using FIGAERO-CIMS and an
Extractive ElectroSpray Ionization time-of-flight mass spectrometer (EESI-ToF-MS). The detailed speciation
depends on analytical method to some extent, though. Moreover, the HOM composition in the particle-phase was
found to depend on aging time and reaction conditions such as dark versus light (Bell et al., 2021; Wu et al., 2021a).

Among the monoterpenes, understanding the reaction system of limonene with $NO_3$ is of specific importance.

The emission of limonene makes the 4[th] largest contribution with an estimated global emission of 11.4 Tg annually,
preceded only by α-pinene, trans-β-ocimene and β-pinene (Guenther et al., 2012). Besides its biogenic origin,
limonene is also a common additive in cleaning products (Nazaroff and Weschler, 2004) and can even be used as a
tracer for fragrances in some places (Gkatzelis et al., 2021). Several studies have shown adverse health effects due
to indoor pollution caused by the ozonolysis of limonene (Clausen et al., 2001; Fan et al., 2003; Carslaw et al., 2012;
Pagonis et al., 2019). Moreover, limonene stands out with its high reactivity towards the $NO_3$ radical (with a lifetime
of 3 min at 298 K at 20 pptv $NO_3$) (Ziemann and Atkinson, 2012), and $NO_3$ oxidation of limonene has high SOA
yield (SOA mass yield 15 to 231 %) (Hallquist et al., 1999; Spittler et al., 2006; Fry et al., 2011; Fry et al., 2014;
Boyd et al., 2017; Berkemeier et al., 2020; Mutzel et al., 2021). A number of earlier studies have provided valuable
insights into the reaction of limonene with $NO_3$ regarding its main products and their formation pathways, the SOA
yield, and the SOA physicochemical properties. For example, Hallquist et al. (1999) measured the SOA mass yield
and revealed the dominance of organic nitrates (ON) and carbonyl compounds in the products. Fry et al. (2011)
determined the organic nitrate yield and proposed a reaction scheme leading to the formation of ON and carbonyls,
and Fry et al. (2014) compared the SOA and ON yields from the $NO_3$ oxidation of α-pinene, β-pinene, and limonene,
and demonstrated why limonene + $NO_3$ leads to more SOA and ON than α-pinene from a structural perspective. Boyd
et al. (2017) found a higher N:C ratio for limonene + $NO_3$ SOA than for β-pinene + $NO_3$ SOA. Finally, Peng et al.
(2018) studied the optical properties of the limonene + $NO_3$ SOA.

Regarding the HOM formation in the reaction of limonene with $NO_3$, Faxon et al. (2018) reported a series of

HOM in the particle phase, including $C_{7-10}$ monomers with 3-11 oxygen atoms and $C_{11-20}$ dimers with 5-19 oxygen
atoms using $I^-$-FIGAERO HR-ToF-CIMS. However, identification of gas-phase HOM products in the limonene +
$NO_3$ reaction is still lacking and their formation mechanisms remain unclear. Theoretical investigations have revealed
that $NO_3$ addition on the endocyclic C=C double bond is more favorable than the exocyclic one due to a lower energy
barrier  (Jiang et al., 2009), and this endocyclic double bond of limonene thus tends to be attacked by $NO_3$ and leads
to products including hydroxy-substituted ON or diketone products. The remaining exocyclic double bond can also
be attacked by $NO_3$ in secondary chemistry, leading to more functionalized products (Fry et al., 2011).

The formation of HOM via autoxidation involves a sequence of multiple intramolecular H-shift and $O_2$ addition

reactions, and results in highly oxygenated peroxy radicals (HOM-$RO_2$•) (Ehn et al., 2014). These HOM-$RO_2$• can
react similarly to traditional $RO_2$• (Bianchi et al., 2019). The bimolecular reactions of HOM-$RO_2$• with $RO_2$•, $HO_2$•
and NO lead to highly oxidized closed shell products including carbonyls, hydroperoxides, alcohols, or organic
nitrates as termination groups (R1 to R3), or form accretion products (R4) (Ehn et al., 2014; Mentel et al., 2015).
Unimolecular termination reactions of HOM-$RO_2$• lead to carbonyls or epoxides (R5 to R6) (Crounse et al., 2013).
On the other hand, reactions of HOM-$RO_2$• with NO, $RO_2$•, $NO_3$ at nighttime can lead to alkoxy radicals as chain
propagating steps (R7 to R9):

$$RO_2\bullet + RO_2\bullet \rightarrow R_HC=O + ROH + O_2 \qquad (R1)$$

$$RO_2\bullet + HO_2\bullet \rightarrow ROOH + O_2 \qquad (R2)$$

$$RO_2\bullet + NO \rightarrow RONO_2 \qquad (R3)$$

$$RO_2\bullet + RO_2\bullet \rightarrow ROOR + O_2 \qquad (R4)$$

$$RO_2\bullet \rightarrow R=O + OH \qquad (R5)$$

$$RO_2\bullet \rightarrow epoxide + OH \qquad (R6)$$

$$RO_2\bullet + NO \rightarrow RO\bullet + NO_2 \qquad (R7)$$

$$RO_2\bullet + RO_2\bullet \rightarrow RO\bullet + RO\bullet \qquad (R8)$$

$$RO_2\bullet + NO_3 \rightarrow RO\bullet + NO_2 + O_2 \qquad (R9)$$

If the reactive HOM-RO• products undergo an H-migration reaction, they will again form HOM-RO$_2$ radicals
("alkoxy-peroxy" pathway) (Mentel et al., 2015), continuing the autoxidation chain. Finally, the HOM-RO• may also
fragment leading to small RO$_2$ radicals, isomerize leading to carbonyls (Bianchi et al., 2019) or react with O$_2$ to form
carbonyls (Ziemann and Atkinson, 2012).

In this study, HOM formation in the NO$_3$ oxidation of limonene was investigated. We report the identification

of gas-phase HOM products, including monomers, dimers and trimers. The formation pathways of dominant products
in each category are proposed based on their time profiles in response to multiple additions of limonene in the
experiment and on the information in literature. Based on this analysis, we estimated HOM yields and discuss the
role of HOM in nucleation and growth of SOA particles.

**2 Experimental and Methods**
**2.1 Experimental setup**
The limonene + NO$_3$ experiment was performed in the atmospheric simulation chamber SAPHIR (Simulation of
Atmospheric PHotochemistry In a large Reaction chamber) at the Forschungszentrum Jülich, Germany. SAPHIR is
a 270 m$^3$ double-wall cylindrical Teflon chamber with a surface-to-volume ratio of ~1 m$^2$ m$^{-3}$. Details of SAPHIR
have been described before (Rohrer et al., 2005; Zhao et al., 2015a; Zhao et al., 2015b; Zhao et al., 2018) and are
only summarized here. Detailed experimental procedures can be found in Fig. 1a. Before each experiment, SAPHIR
was flushed for about 4 h at a flow rate of 370 m$^3$ h$^{-1}$ with high-purity synthetic air (purity >99.9999 % O$_2$ and N$_2$)
in order to clean the chamber. To simulate nighttime conditions for the NO$_3$ chemistry the chamber roof remained
closed throughout the experiment. The experiment was performed under dry conditions (RH <2 %) at a temperature
of 302 ± 3 K. No seed aerosols were used in the experiments. A fan was used for active mixing in the chamber,
leading to a typical mixing time of ~1 min (Fuchs et al., 2013).

NO$_3$ radicals were generated via the reaction of ozone with nitrogen dioxide:

$$NO_2 + O_3 \rightarrow NO_3 + O_2 \qquad (R10)$$

$$NO_2 + NO_3 \leftrightarrow N_2O_5 \qquad (R11)$$

Therefore, $O_3$ and $NO_2$ were first added to the chamber to form $N_2O_5$ and $NO_3$ with mixing ratios of ~2 ppbv
and ~0.15 ppbv, respectively. About 20 min later, 5 ppbv of limonene was added to start the organic chemistry. Five
more additions of limonene followed, with added concentrations of about 3 ppbv, 3 ppbv, 2 ppbv, 2 ppbv, and finally
8 ppbv, which divided the experiment into six periods (P1 to P6) (Fig. 1a). For period P3 and P5, $NO_2$ and $O_3$ were
also added to compensate for the loss of $NO_3$ and $N_2O_5$ (Fig. 1a). The concentrations of $NO_2$ and $O_3$ were maintained
around 20 to 70 ppbv throughout the experiment, ensuring the major loss of limonene was by reaction with $NO_3$
rather than with $O_3$ (Fig. S1 in the SI). In the first ten min of reaction (named period P1a hereafter, Fig. 1a), $NO_3$
accounted for 86 % of the chemical loss of limonene.

**2.2 Instrumentation**

Gas-phase HOM were detected by a Chemical Ionization time-of-flight Mass Spectrometer (CI-APi-TOF, Aerodyne
Research Inc., USA) with a resolution $(m/z)/(\Delta m/z)$ of ~3800 using $^{15}NO_3^-$ as the reagent ion, which is capable of
detecting organic molecules with high oxygen content (Eisele and Tanner, 1993; Jokinen et al., 2012). The mass
spectra were analyzed using the software Tofware (Tofwerk/Aerodyne) in Igor Pro (WaveMetrics, Inc.). Peak
identification was conducted by a high-resolution analysis (examples shown in Fig. S2). We observed several peaks
which were obviously products from the isoprene + $NO_3$ reaction, such as $C_5H_{10}N_2O_8 \cdot {}^{15}NO_3^-$ at m/z 289. Such peaks
were present before the limonene oxidation reaction started, suggesting that these compounds preexisted in the
chamber. These isoprene oxidation products were likely formed in an isoprene + $NO_3$ experiment performed two
days before (Zhao et al., 2021) and released slowly from chamber walls due to their semi-volatile character. Their
total concentration is less than 1 ppt. All the isoprene-HOM observed ($C_5H_9NO_{7,10}$, $C_5H_8N_2O_{8-10}$, $C_5H_{10}N_2O_8$,
$C_5H_9N_3O_{9,10}$) are saturated and do not contain C=C double bond. The isoprene-HOM will not influence the reaction
of limonene with $NO_3$ in this study. Therefore, they are not discussed as products from the limonene oxidation in our
experiment. (However, we cannot exclude that they were partly generated from fragmentation in the limonene + $NO_3$
reaction.)
A set of instruments were used to measure other gas-phase species, including VOC, $NO_x$, $O_3$, $NO_3$ and $N_2O_5$
(Shen et al., 2021). Concentrations of $NO_3$ and $N_2O_5$ were measured in-situ using a home-built diode laser-based,
cavity ring-down spectrometer similar to the instrument described in the work by Wagner et al. (2011). The
concentrations of limonene were measured using a Proton Transfer Reaction Time-of-Flight Mass Spectrometer
(PTR-TOF-MS, Ionicon Analytik, Austria). The SOA number concentration, surface concentration and size
distribution were detected by an SMPS (TSI DMA3081/TSI CPC3786) and a CPC (TSI 3785). Temperature and
relative humidity were continuously monitored throughout the experiment.

**2.3 Determination of HOM concentration and "primary" HOM yield**

HOM concentrations were obtained from the normalized signals to the total signals of the mass spectra (nc,
normalized counts) by applying a calibration coefficient (C) of $2.5\times10^{10}$ molecule $cm^{-3}$ $nc^{-1}$. C was determined using
$H_2SO_4$ as the charging efficiency of HOM and $H_2SO_4$ are considered to be equal (Ehn et al., 2014; Pullinen et al.,
2020; Shen et al., 2021). The details of determination of the calibration coefficient are shown in the supplement S1.

A mass-independent transmission efficiency was used according to our previous study, which causes an additional uncertainty of 14 % (Pullinen et al., 2020). In this previous study, the transmission efficiency curve of nitrate CI-APi-TOF was determined and found to monotonously decrease with increasing mass of ions but only slightly depend on the mass range (14 % change). As we used the same setting as our previous study, we have included the slight dependence of transmission on m/z in the uncertainties. The concentrations of HOM were corrected for chamber wall losses, which were determined for a number of HOM similar to our previous study (Zhao et al., 2018), with details described in the supplement. When the chamber is actively mixed, the wall loss was determined to be $(2.2\pm0.2)\times10^{-3}$ $s^{-1}$. As the HOM yield was determined during the first 3 min of the experiment, we considered the wall loss rate to be constant ($2.2\times10^{-3}$ $s^{-1}$) during this period. Sensitivity analysis showed that the HOM yield in this study is not very sensitive to the wall loss rate and is changing by only +0.88 % and -0.44 % if the wall loss rate is varied by +100 % or -50 %.

The HOM yield was calculated as:

$$Y = \frac{[HOM]}{[VOC]_r} = \frac{I(HOM)\cdot C}{[N_2O_5]_r}$$
(Eq. 1)

where $[HOM]$ is the concentration of HOM, $I(HOM)$ is the total signal intensity of HOM, C is the calibration factor, and $[VOC]_r$ and $[N_2O_5]_r$ stand for the concentrations of limonene and $N_2O_5$ reacted, respectively. We used the reacted concentration of $N_2O_5$ rather than the measured reacted limonene concentration as a large fraction of limonene was already reacting away during the VOC injection before it was homogeneously mixed in the chamber. During this part of the experiment, the high limonene concentration resulted in a rapid loss of $NO_3$, such that every $NO_3$ formed from the decomposition of $N_2O_5$ reacted with limonene:

$$N_2O_5 \rightarrow NO_2 + NO_3$$
(R12)

$$\text{limonene} + NO_3 \rightarrow \text{Products}$$
(R13)

The initial $NO_3$ concentration before the limonene injection was small compared to the time-integrated loss of $N_2O_5$, and other $NO_3$ loss processes were negligible right after the limonene injection, so that the observed decrease in the $N_2O_5$ concentration equals indeed the consumption of limonene. The wall loss rate constant of $N_2O_5$ in the SAPHIR chamber is $7.2\times10^{-5}$ $s^{-1}$ (Fry et al., 2009). As the HOM yield determination is based on the first 3 min, the wall loss of $N_2O_5$ can be ignored compared to the loss via the reaction of $NO_3$ with limonene.

The uncertainty of the HOM yield was estimated to be -55 %/ +117 % based on the combined uncertainties of the HOM-ON peak intensities (~10 %), the limonene concentration (~15 %), the transmission efficiency (-0 %/+14 %) and the calibration factor (-52 %/ +101 %) using error propagation (Zhao et al., 2021). The first 3 min after the injection of limonene were used to calculate the HOM yield, when most of the first-generation oxidation products were produced and negligible particles were yet formed. The HOM yield thus reflects the "primary" HOM yield.

**2.4 Determination of HOM condensation on SOA**

The SOA mass from the condensation of HOM was calculated to evaluate the role of HOM for the SOA mass growth. Detailed estimation methods are described in the supplement, including the determination of particle wall loss and dilution loss rate (Sect. S2). In brief, the growth rate of SOA through HOM vapor condensation is based on the

collision rate of vapor molecules with aerosols in the kinetic regime. The Fuchs-Sutugin approach is applied to describe the correction for transition from the kinetic to the diffusion regime (Fuchs and Sutugin, 1971; Ehn et al., 2014). Based on the volatility of HOM, we considered two scenarios for HOM condensation. In Scenario 1, all HOM were assumed to irreversibly condense on the surface of particles leading to particle mass growth. In Scenario 2, only the irreversible uptake of LVOC and ULVOC/ELVOC compounds were considered to contribute to the growth of SOA particles in order to examine the role of LVOC and ELVOC while IVOC and SVOC were not included, although they may also contribute to SOA. The calculation of saturation concentration $C^*$ (in μg/m$^3$) of each HOM was done based on their molecular compositions using two different parameterizations considering the uncertainties in estimating volatility (Wu et al., 2021b):

1. an updated version of the parameterization of Donahue et al. (2011) by Mohr et al. (2019) (Scenario 2a):

$$log_{10}(C) = (25 - n_C) \times 0.475 - (n_O - 3n_N) \times 0.2 - 2\frac{(n_O - 3n_N)n_C}{(n_C + n_O - 3n_N)} \times 0.9 - n_N \times 2.5 \quad \text{(Eq. 2)}$$

where $n_C$, $n_O$, $n_N$ and $n_H$ are the number of carbon, oxygen, nitrogen and hydrogen atoms of the compound, respectively.

2. a parameterization based on HOM from α-pinene ozonolysis by Peräkylä et al. (2020) (Scenario 2b):

$$log_{10}(C) = n_C \times 0.18 - n_H \times 0.14 - n_O \times 0.38 + n_N \times 0.80 + 3.1 \quad \text{(Eq. 3)}$$

with similar parameter notation.

**2.5 Simulations of the RO$_2$• loss pathway based on the Master Chemical Mechanism (MCM)**

The RO$_2$• loss pathways were estimated based on MCM simulations (http://mcm.york.ac.uk/). The gas-phase reactions of limonene + NO$_3$ under dark condition were simulated using iChamber, an open-source program (https://sites.google.com/view/wangsiyuan/models?authuser=0) (Wang and Pratt, 2017). The default chemistry of limonene + NO$_3$ in the MCM was applied in this study (Saunders et al., 2003b). Photolysis reactions were excluded by setting the zenith angle to 90º. Concentrations of O$_3$, NO$_3$, NO$_2$ and N$_2$O$_5$ as well as temperature and relative humidity were constrained to the experimental data with a time resolution of 1 min. The chamber dilution rate of $1.5 \times 10^{-5}$ s$^{-1}$ was applied to all species. The P1 period was simulated using the above conditions and the initial concentrations of limonene were added in the model according to the experimental procedures. The sum of all 140 RO$_2$• in the limonene subset of MCM v3.3.1 were used in the usual way to estimate the loss rates of RO$_2$• bimolecular reactions. The reaction rate constants are provided in Table S3, and calculated loss rates are shown in Fig. S3. We note that the MCM reaction schemes do not include the accretion reactions between HOM-RO$_2$•. Berndt et al. (2018a) determined the rate constant of accretion reaction of C$_{10}$H$_{15}$O$_4$• formed via α-pinene ozonolysis to be ~$1 \times 10^{-11}$ cm$^3$ molecule$^{-1}$ s$^{-1}$, which is of the same order as the upper limit for RO$_2$• + RO$_2$• reactions used in the MCM schemes for functionalized peroxy radicals such as acyl peroxy radicals (Jenkin et al., 1997; Saunders et al., 2003a). However, currently we do not see a reliable updated set of rate coefficients that are applicable to the reaction system in this study. If the rate constants of some RO$_2$• + RO$_2$• reactions were higher than those used in MCM, the concentrations of RO$_2$• would be lower and relative importance of RO$_2$• + RO$_2$• in RO$_2$• fate would increase. Several simulation results are shown in Fig. S4, including NO$_3$, N$_2$O$_5$, limonene, RO$_2$•, reaction rate of limonene with NO$_3$

($k\times$limonene$\times NO_3$), and examples of 1st and 2nd-generation $RO_2\bullet$.

In the early stage of each period, $RO_2\bullet$ mainly reacted with $RO_2\bullet$ and $NO_3$, although in the later stage the reaction

with $NO_2$ also contributed to a significant fraction of $RO_2\bullet$ loss (Fig. S3, showing period P1 as an example). During
the period P1a when our peak assignment was based on, the $RO_2\bullet$ loss was dominated by $RO_2\bullet$ + $RO_2\bullet$ and $RO_2\bullet$ +
$NO_3$.

**3 Results and discussion**
**3.1 Experiment overview and observed HOM**
After each limonene addition, the concentration of limonene rose first and then rapidly declined, while the
concentrations of $NO_3$ and $N_2O_5$ rapidly decreased due to the fast reaction between limonene and $NO_3$ and gradually
increased when limonene had been consumed (Fig. 1a). About 10 min after the first limonene addition, new particles
had already formed and quickly grew in size (Fig. 1b). Therefore, we used the first 10 min reaction time (period P1a)
to identify gas-phase HOM products, and the whole experiment to examine the contribution of HOM to SOA.

During period P1, HOM were quickly formed. We identified about 280 HOM compounds, including monomers

($C_7$-$C_{10}$, ~280-460 Th), dimers ($C_{17}$-$C_{20}$, ~490-700 Th), and trimers ($C_{26}$-$C_{30}$, ~720-960 Th) (Fig. 2a). Their detailed
formulas can be found in Table S1. HOM on the horizontal lines of the Kendrick mass defect plot (O-based) (Fig. 3
and Fig S5, S6) share the same number of C, N and H atoms, with the number of oxygen atoms increasing from left
to right. Such HOM compounds are defined as a family. We notice that most monomer peroxy radical families are
each related to two monomer closed-shell product families, with one H atom more or one H atom less, which are the
expected termination products of $RO_2\bullet$ + $RO_2\bullet$ reactions, or if $HO_2\bullet$ is present, $RO_2\bullet$ + $HO_2\bullet$ termination products.
These three related families are defined as a "series", with the same number of C and N number, such as $C_{10}H_{15\text{-}17}NO_{6\text{-}14}$.
In total, we identified 6 monomer series ($C_{10}H_{15\text{-}17}NO_{6\text{-}14}$, $C_{10}H_{14\text{-}16}N_2O_{9\text{-}15}$, $C_{10}H_{14\text{-}16}O_{7\text{-}12}$, $C_9H_{13\text{-}15}NO_{7\text{-}14}$,
$C_8H_{11\text{-}13}NO_{6\text{-}13}$ and $C_7H_{9\text{-}11}NO_{7\text{-}11}$) and 1 monomer family ($C_{10}H_{17}N_3O_{12\text{-}16}$), 11 dimer families ($C_{20}H_{31}NO_{10\text{-}15}$,
$C_{20}H_{33}NO_{12\text{-}16}$, $C_{20}H_{32}N_2O_{9\text{-}20}$, $C_{20}H_{31}N_3O_{14\text{-}20}$, $C_{20}H_{33}N_3O_{12\text{-}20}$, $C_{20}H_{34}N_4O_{15\text{-}20}$, $C_{20}H_{32}O_{13\text{-}16}$, $C_{19}H_{29}NO_{10\text{-}13}$,
$C_{19}H_{31}NO_{10\text{-}15}$, $C_{19}H_{30}N_2O_{10\text{-}18}$ and $C_{19}H_{31}N_3O_{15\text{-}19}$), and 3 trimer families ($C_{30}H_{47}N_3O_{18\text{-}24}$, $C_{30}H_{48}N_4O_{16\text{-}24}$ and
$C_{29}H_{46}N_4O_{19\text{-}24}$). The monomer family $C_{10}H_{17}N_3O_{12\text{-}16}$ are not classified as series because the supposedly relevant
families are not clearly identified. Compounds containing at least one nitrogen atom accounted for more than 90 %
of the identified HOM products. We assume that compounds containing nitrogen atoms are organic nitrates, because
other N-containing species such as amines or nitro compounds are very unlikely to be formed from the reaction of
limonene with $NO_3$. Organic nitrates formed in this study could be alkylnitrates, or (acyl)peroxynitrates formed via
the reaction of $RO_2\bullet$ with $NO_2$.

During period P1a, in the absence of particles, both HOM monomers and oligomers were observed, including

monomers (47 %), dimers (47 %) and trimers (6 %) (Fig. 2a). Concentrations of gas-phase dimers and trimers
decreased evidently after particle formation (Fig. 2b, 5, 6), indicating a fast gas-particle condensation and strong
tendency of oligomers to condense on particles.

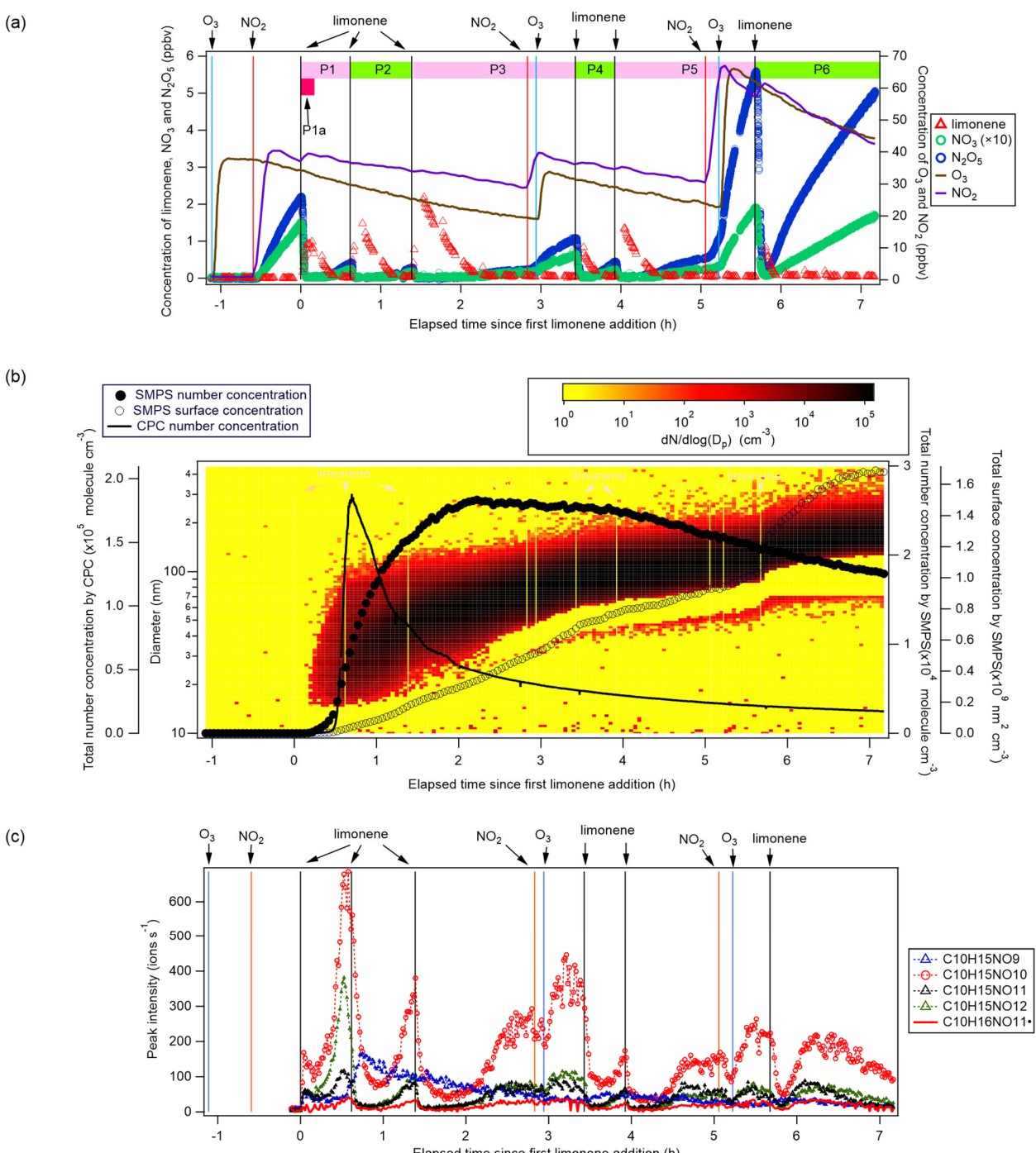


Figure 1. (a) Time series of the concentrations of limonene, $NO_3$, $N_2O_5$ (left panel, the concentrations of $NO_3$ are magnified by a factor of 10 for clarity), as well as $O_3$ and $NO_2$ (right panel). The time periods P1 to P6 as well as P1a are also marked. (b) Total particle concentration and its size distribution during the whole period of experiment detected by SMPS and CPC. The solid and hollow black circles refer to total number concentration and total surface concentration detected by SMPS, respectively. Colors represent particle number concentration distribution based on $\log(D_p)$. The solid black line refers to total number concentration detected by CPC. (c) Time series of peak intensity of typical products of the $C_{10}H_{15}NO_x$ family and $C_{10}H_{16}NO_{11}\bullet$ as a representative of the $C_{10}H_{16}NO_x\bullet$

family. Vertical lines indicate the time of $O_3$ and $NO_2$ additions, as well as six limonene injections.

Based on their typical time series (Fig. 1c), products can be classified as first-generation or second-generation
products. Generally, the concentrations of first-generation products, which result from the direct reaction of limonene
with $NO_3$, are expected to quickly increase after the limonene addition, followed by a steady decline due to wall loss
or chemical reactions. Concentrations of typical second-generation products, which result from further reactions of
first-generation products, are expected to show a gradually increasing concentration pattern after a limonene addition
and reach their maximum concentration later than first-generation products. These general expectations are modified
in our case, since the particle concentration increased in our experiment (Fig. 1b) and the condensational sink of
HOM products became stronger over time. Thus, an increase in concentration suggests that the increasing
condensational sink was exceeded by increasing production with time, i.e. from second-generation pathways.
To sum up, gas-phase HOM formed in the limonene + $NO_3$ system were dominated by HOM monomers and
dimers. Time series patterns of the products indicate multiple generations of reaction pathways.

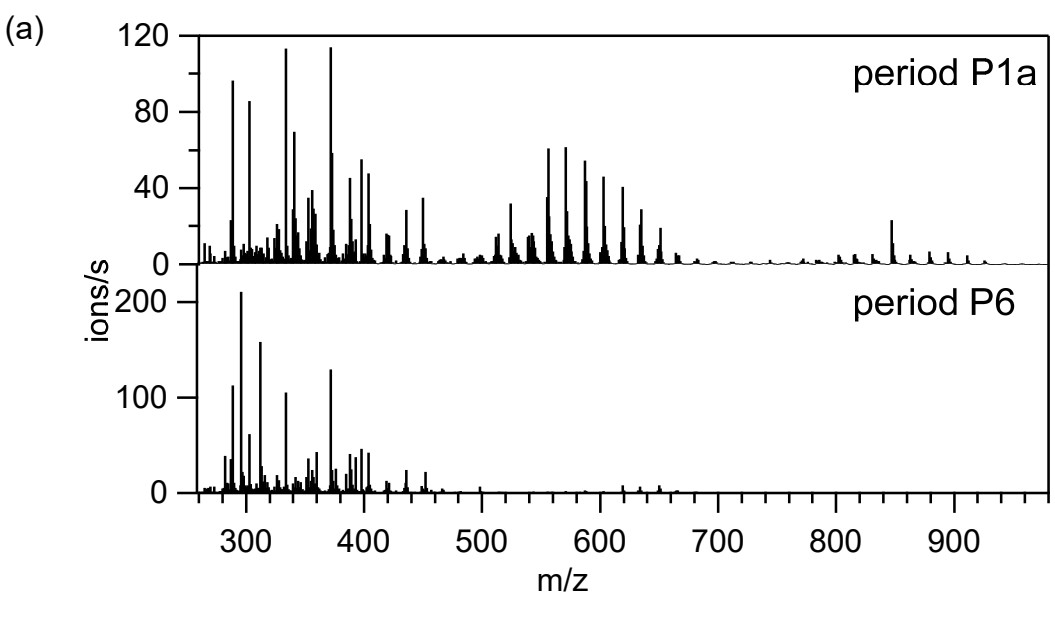

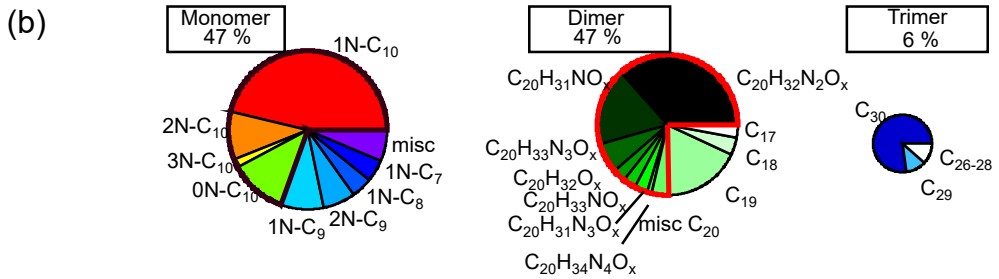


Figure 2. (a) Average mass spectra of the first 10 min reaction time after the first addition of limonene (period P1a,
upper panel) and the last limonene addition period till particles reached maximum mass concentration (period P6,
lower panel). (b) Pie charts (from left to right: rainbow, green, and blue colors) representing the relative contributions
of identified families to HOM monomers, dimers, and trimers, respectively, during the P1a period. The area of each
pie is in proportion to their concentrations during the P1a period.

### 3.2 Monomers and their formation pathways

### 3.2.1 Overview of HOM monomers

A number of HOM monomer families were detected with an increasing oxygenation pattern at 16 Th intervals (Fig.
3). Such a pattern is attributed to autoxidation of $RO_2\bullet$ (with 32 Th interval for each $O_2$ addition) plus the alkoxy-
peroxy pathway (shifted by 16 Th compared with exclusive autoxidation) as discussed below. During period P1a, the
most abundant HOM monomers are $C_{10}$ compounds (64 %), such as peroxy radicals $C_{10}H_{16}NO_x\bullet$ and closed-shell
products $C_{10}H_{15}NO_x$ and $C_{10}H_{17}NO_x$, which are carbonyl compounds and hydroxyl or hydroperoxy compounds from
the termination reactions of $C_{10}H_{16}NO_x\bullet$, respectively (R14).
$$C_{10}H_{16}NO_x\bullet + C_{10}H_{16}NO_x\bullet \rightarrow C_{10}H_{15}NO_x + C_{10}H_{17}NO_x \qquad (R14)$$
According to the nitrogen atoms contained, $C_{10}$-HOM monomers can be classified into 1N-, 2N-, 3N-monomers,
and monomers without nitrogen atoms. While 1N-$C_{10}$ HOM monomers were likely formed by direct $NO_3$ addition
to limonene, $C_{10}$ HOM monomers containing multiple N atoms were likely formed via multiple reaction steps.
Besides $C_{10}$ HOM monomers, $C_{6-9}$ HOM monomers were also observed. These $C_{6-10}$ families are discussed below in
the order of their contributions to HOM monomers.

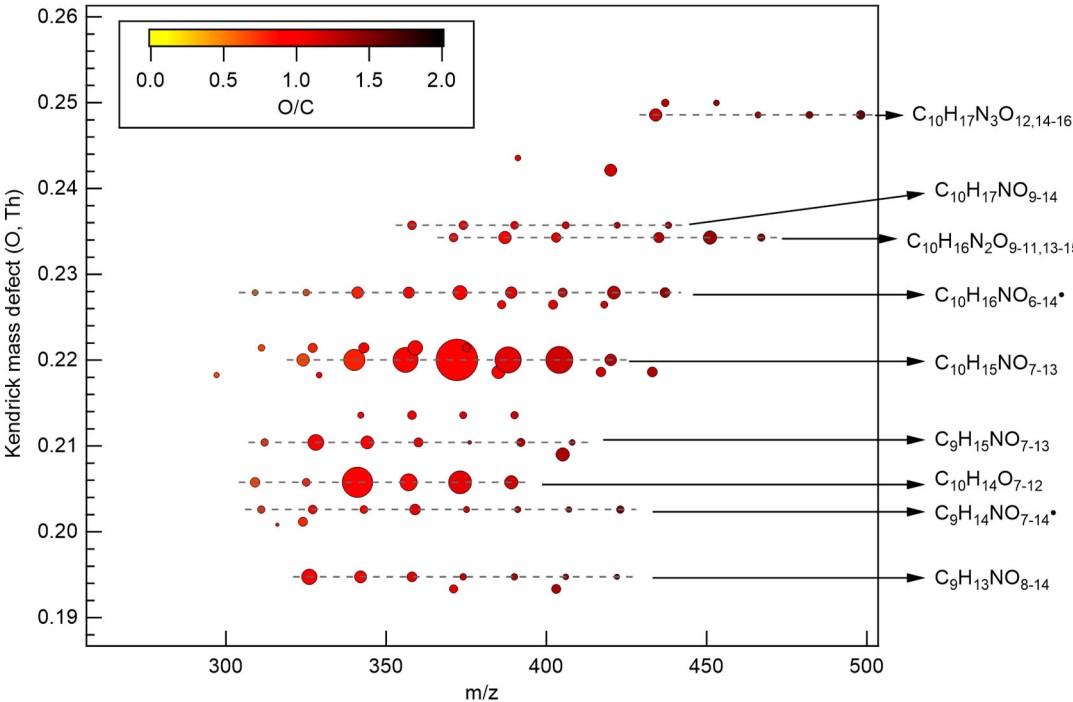


Figure 3. Kendrick mass defect plot (O-atom-based) of major monomer products. The area of the circles is
proportional to the average intensity of each compound during the P1a period with the largest circle representing
$C_{10}H_{15}NO_{10}$. The color denotes O/C ratios. Dashed lines indicate major product families. For clarity, the reagent ions
[15]NO$_3^-$ is omitted from molecular formula. The calculation of O-atom-based Kendrick mass defect includes two steps.
First, the IUPAC mass scale (based on the [12]C atomic mass as exactly 12 Da) is rescaled to Kendrick mass: Kendrick
mass = IUPAC mass × (16/15.9949), which converts the mass of O from 15.9949 to exactly 16. Then, Kendrick mass
defect is given by: Kendrick mass defect = nominal Kendrick mass – exact Kendrick mass. Thus, compounds with
the same number of each kind of atom except for O have equal O-atom-based Kendrick mass defect, and are shown
in a horizontal line in the O-atom-based Kendrick mass defect plot.

**3.2.2 1N-C$_{10}$ monomers**
Among C$_{10}$ HOM monomers, the 1N-C$_{10}$ families were most abundant and included stable closed-shell products
C$_{10}$H$_{15}$NO$_x$ (x=7-13) and C$_{10}$H$_{17}$NO$_x$ (x=9-14) and peroxy radicals C$_{10}$H$_{16}$NO$_x$• (x=6-14). The concentration of
C$_{10}$H$_{16}$NO$_{11}$• increased in the later phase of each limonene addition period (Fig. 1c), showing mostly a time profile
of a second-generation product, similar as most of the other radicals in the C$_{10}$H$_{16}$NO$_x$• family (Fig. S7). However,
the time series of C$_{10}$H$_{15}$NO$_x$ compounds showed an overlaying pattern of first- and second-generation products
dominated by a second-generation time profile with the exception of C$_{10}$H$_{15}$NO$_9$ (Fig. 1c). Due to sensitivity
restrictions of CIMS, the primary peroxy radical C$_{10}$H$_{16}$NO$_5$• was not detected, which was supposed to show 1[st]-
generation pattern. The absence of first-generation characteristics of the time profile of most HOM peroxy radicals
C$_{10}$H$_{16}$NO$_{x(x\geq6)}$• may be attributed to two possible reasons. They either did not undergo efficient autoxidation, or they
underwent immediate conversion including autoxidation and/or bimolecular reactions with other RO$_2$• or NO$_3$
forming closed-shell products such as dimers or continuing the radical chain forming RO•. The instantaneous increase
of 2N-dimers and trimers after the first limonene addition shown below suggests that C$_{10}$H$_{16}$NO$_{x(x\geq6)}$• were indeed
formed efficiently via autoxidation. Therefore, the latter reason is more likely. At this time, we do not have a
reasonable explanation for the trend of C$_{10}$H$_{15}$NO$_9$, though we should consider that there are many isomers at play,
which may have very different chemical pathways (un)available.
Since the C$_{10}$H$_{15}$NO$_x$ family showed an overlaying pattern of the first-generation and second-generation
products, they likely contained multiple isobaric substances produced through different pathways. Based on the
literature, possible formation pathways of these products were tentatively proposed (Seinfeld and Pandis, 2006;
Vereecken and Peeters, 2010; Mentel et al., 2015; Vereecken and Nozière, 2020). As an example of the pathways to
form first-generation products, C$_{10}$H$_{16}$NO$_{2x-1}$• (with an odd number of oxygen atom) and their corresponding
termination products can be formed via autoxidation of the first peroxy radical C$_{10}$H$_{16}$NO$_5$• (R1OO), showing
C$_{10}$H$_{16}$NO$_9$• (R3OO) as an example (Scheme 1a, first-generation products). C$_{10}$H$_{16}$NO$_{2x}$• (with an even number of
oxygen atom) can be formed via alkoxy-peroxy channels. For example, the ring-opening of the alkoxy radical
C$_{10}$H$_{16}$NO$_4$• (R1O), which was formed via the reaction of C$_{10}$H$_{16}$NO$_5$• (R1OO) with another RO$_2$• or NO$_3$ radical
(Scheme 1a, first-generation products). Ring-opening of R1O leads to C$_{10}$H$_{16}$NO$_6$• (R4OO), which can undergo
autoxidation forming C$_{10}$H$_{16}$NO$_{2x}$•. In addition, the alkoxy radical C$_{10}$H$_{16}$NO$_4$• (R1O) is susceptible to ring-opening
reactions (Novelli et al., 2021), which can lead to a first-generation stable product 3-isopropenyl-6-oxoheptanal

(endolim, TP1) after C-C bond cleavage followed by the elimination of a $NO_2$ fragment (Scheme 1b, second-generation products). Endolim (TP1) has been detected as a major product in previous limonene + $NO_3$ studies (Hallquist et al., 1999; Spittler et al., 2006).

As an example of second-generation chemistry, the remaining double bond of endolim could react with $NO_3$ to form $RO_2\bullet$, followed by the autoxidation to form second-generation $C_{10}H_{16}NO_x\bullet$ (with odd number of oxygen atoms). Similar to first-generation pathways, second-generation $C_{10}H_{16}NO_x\bullet$ with even number of oxygen atoms can be formed via alkoxy-peroxy channel. From the time profile of $C_{10}H_{15}NO_x$, the second-generation pathway (Scheme 1b) was expected to play a more important role, in agreement with the theoretical result by Kurtén et al. (2017), in which the two bond-cleavage pathways of limonene-derived $RO\bullet$ radical were considered. It is worth mentioning that the reaction products of limonene with $O_3$ may also react with $NO_3$, forming $C_{10}H_{16}NO_x\bullet$ (Scheme S1). However, as shown above, this was a minor pathway in our experiment (Sect. 2.1). We would like to note that to simplify the scheme, only the reaction of $NO_3$ with the endocyclic double bond is presented, since this reaction is faster than that with the exocyclic double bond (Jiang et al., 2009; Fry et al., 2011).

$C_{10}H_{16}NO_x\bullet$ with both even and odd number of oxygen atoms as well as their termination products had comparable abundance, which suggests that the alkoxy-peroxy pathway was important for $RO_2\bullet$ formation in this reaction. This finding is analogous to the findings in the reaction of a number of alkenes with $O_3$ and in the reaction of isoprene and β-pinene with $NO_3$ (Mentel et al., 2015; Zhao et al., 2021; Shen et al., 2021).

Scheme 1. Illustrative scheme for HOM formation in the limonene + $NO_3$ reaction. (a) Example formation pathways leading to first-generation $1N-C_{10}$ HOM-$RO_2$ radicals ($C_{10}H_{16}NO_x\bullet$ with even or odd numbers of O-atoms). (b) Second-generation scheme involving the formation of endolim. (c) Scheme of intramolecular termination of $RO_2\bullet$

radicals forming carbonyl products taking the $C_{10}H_{16}NO_9\bullet$ radical as an example. Note that the depicted reactions
may not be the dominant pathways.

Among 1N-$C_{10}$ monomers, concentrations of carbonyl compounds were much higher than the sum of hydroxy-

and hydroperoxy-substituted compounds (Table 1). According to Hyttinen et al. (2015), for nitrate CI-APi-TOF,
HOM containing two hydrogen bond donors (such as -OOH and -OH group) have strong binding energy with $NO_3^-$.
Additional hydrogen bond donors only enhance the binding energy marginally. If we compare HOM carbonyl product
(such as $C_{10}H_{15}NO_{10}$) with the corresponding hydroxy product ($C_{10}H_{17}NO_{10}$), they only differ in one functional group.
As both are highly functionalized, it is likely that HOM carbonyl have a quite similar sensitivity with HOM alcohol.
If the sensitivity of carbonyl HOM were lower, this would result in even more dominance of carbonyl HOM over
hydroxyl HOM. Thus, we conclude that carbonylnitrates are more abundant than hydroxynitrates or
hydroperoxynitrates. This finding is likely attributed to unimolecular termination reactions of $RO_2\bullet$, although reaction
paths via $RO\bullet$ also cannot be excluded. Smaller unbranched $RO\bullet$ tend to react with $O_2$ forming carbonyl compounds
while for larger or branched $RO\bullet$, isomerization can also form carbonyl compounds and is a more energetically
favorable and thus faster pathway compared with the reaction with $O_2$ (Ziemann and Atkinson, 2012). The importance
of unimolecular termination reactions of HOM-$RO_2\bullet$ and the resulting high ratio of carbonyl compounds to
hydroxyl/hydroperoxyl compounds has also been found in the reaction system of β-pinene + $NO_3$ (Shen et al., 2021;
Dam et al., 2022). This high ratio is also consistent with findings in the ozonolysis of alkenes (Mentel et al., 2015),
where unimolecular termination reactions were also proposed to be the likely explanation (Crounse et al., 2013;
Rissanen et al., 2014). As discussed in our previous study by Shen et al. (2021), this higher abundance of
carbonylnitrates is not likely to be explained by the reaction of alkoxy $RO\bullet + O_2$ forming carbonyls and $HO_2\bullet$,
decomposition of β-nitrooxyperoxynitrate or self-reactions of $RO_2\bullet$ via the Bennett and Summers mechanism forming
carbonyls and $H_2O_2$. Reactions between $RO_2\bullet$ in general should produce overall equal amounts of carbonyl and
hydroxyl compounds. The decomposition of β-nitrooxyperoxynitrate is slow in the gas-phase. The reaction of alkoxy
$RO\bullet$ with $O_2$ for large $RO\bullet$ is generally slower than isomerization and decomposition (Vereecken and Peeters, 2009,
2010). Thus, the higher abundance of carbonylnitrates compared to hydroxynitrates may be attributed to unimolecular
termination of HOM-$RO_2\bullet$. In addition, isomerization of $RO\bullet$ forming carbonyl compounds may also contribute to
this finding. Our result thus further emphasizes that unimolecular termination reactions of $RO_2$ radicals are important
pathways in the formation of HOM monomers derived from the reactions of monoterpenes with $NO_3$ (Shen et al.,
2021). Scheme 1c shows this unimolecular termination process using a $C_{10}H_{16}NO_9\bullet$ radical as an example.
$C_{10}H_{16}NO_9\bullet$ undergoes a 1,4-H-shift and $O_2$ addition to form a $C_{10}H_{16}NO_{11}\bullet$ radical. The $C_{10}H_{16}NO_{11}\bullet$ radical further
undergoes an H-shift of the α-OOH H-atom, which produces a carbonyl closed-shell product as well as an OH•
radical.

For 1N-$C_{10}$ HOM monomers, the products detected in this study generally agree with previous laboratory and

field studies on the reaction of limonene and other monoterpenes. Faxon et al. (2018) also observed $C_{10}H_{15}NO_x$ as

the most prevalent products in the particle phase from limonene + $NO_3$. In the SOAS campaign, both $C_{10}H_{15}NO_x$ and $C_{10}H_{17}NO_x$ products were detected and were believed to be products of nighttime chemistry (Lee et al., 2016). The high abundance of 1N-$C_{10}$ HOM monomers is consistent with the finding that $C_{10}H_{15}NO_x$ and $C_{10}H_{17}NO_x$ dominate the chemical composition of SOA formed via $NO_3$ oxidation of α-pinene and β-pinene, as shown in previous chamber studies (Takeuchi and Ng, 2019).

In summary, 1N-$C_{10}$ HOM monomers are mainly formed via second-generation pathways, and unimolecular termination of $RO_2\bullet$ likely plays an important role leading to higher abundance of carbonyl HOM-ON ($C_{10}H_{15}NO_x$) than hydroxy/hydroperoxy HOM-ON ($C_{10}H_{17}NO_x$).

Table 1. Observed $C_{10}H_{16}NO_x\bullet$ radicals (m) and their termination products, including carbonyl compounds (m-17), hydroxyl compounds (m-15), and hydroperoxy compounds (m+1). Their concentrations during period P1a are normalized to that of $C_{10}H_{15}NO_{10}$, which had the highest concentration among the families of 1N-$C_{10}$ monomers. Their relative signal intensities during the P1a period are shown in the second line of each cell.

| Peroxy radical m | Carbonyl m-17 | Hydroxy m-15 | Hydroperoxy m+1 |
|---|---|---|---|
| $C_{10}H_{16}NO_6\bullet$ 1.5 % | | | |
| $C_{10}H_{16}NO_7\bullet$ 2.0 % | | | |
| $C_{10}H_{16}NO_8\bullet$ 6.7 % | $C_{10}H_{15}NO_7$ 8.0 % | | |
| $C_{10}H_{16}NO_9\bullet$ 6.0 % | $C_{10}H_{15}NO_8$ 25.2 % | | $C_{10}H_{17}NO_9$ 3.7 % |
| $C_{10}H_{16}NO_{10}\bullet$ 10.2 % | $C_{10}H_{15}NO_9$ 34.6 % | $C_{10}H_{17}NO_9$ 3.7 % | $C_{10}H_{17}NO_{10}$ 3.6 % |
| $C_{10}H_{16}NO_{11}\bullet$ 6.6 % | $C_{10}H_{15}NO_{10}$ 100.0 % | $C_{10}H_{17}NO_{10}$ 3.6 % | $C_{10}H_{17}NO_{11}$ 3.0 % |
| $C_{10}H_{16}NO_{12}\bullet$ 4.1 % | $C_{10}H_{15}NO_{11}$ 39.0 % | $C_{10}H_{17}NO_{11}$ 3.0 % | $C_{10}H_{17}NO_{12}$ 2.3 % |
| | $C_{10}H_{15}NO_{12}$ 41.2 % | $C_{10}H_{17}NO_{12}$ 2.3 % | $C_{10}H_{17}NO_{13}$ 1.5 % |
| $C_{10}H_{16}NO_{14}\bullet$ 4.7 % | $C_{10}H_{15}NO_{13}$ 6.7 % | $C_{10}H_{17}NO_{13}$ 1.5 % | $C_{10}H_{17}NO_{14}$ 1.8 % |
| | | $C_{10}H_{17}NO_{14}$ 1.8 % | |

### 3.2.3 2N and 3N-$C_{10}$ monomers

$C_{10}$ monomers with 2 and 3 nitrogen atoms accounted for 27 % and 1 % of HOM monomers, respectively. They were likely formed via the reaction of a second attack of $NO_3$ to the first-generation products as the 1N-$C_{10}$ closed-shell

products formed via the reactions shown in Scheme 1a should contain a remaining limonene C=C double bond.
Typical 2N- and 3N-HOM showed a second-generation time profile (Fig. 4). For clarity, only periods P1 to P3 are
shown. This time profile is consistent with the pathways with multiple $NO_3$ attacks. Scheme 2 shows possible
formation pathways of 2N- and 3N-$C_{10}$ monomers. 2N-$C_{10}$ HOM were likely to be formed from $NO_3$ oxidation of
1N-$C_{10}$ monomers ($C_{10}H_{15}NO_x$ and $C_{10}H_{17}NO_x$), resulting in $C_{10}H_{15}N_2O_x\bullet$ and $C_{10}H_{17}N_2O_x\bullet$ (Scheme 2a, 2b). While
$C_{10}H_{15}N_2O_x\bullet$ (x=9-12) were observed, $C_{10}H_{17}N_2O_x\bullet$ could not be uniquely identified because the peaks of the
$C_{10}H_{17}N_2O_x\bullet$ and $C_{10}H_{15}NO_x$ families are too close in the mass spectra to be separated based on the resolution of our
mass spectrometer. 3N-$C_{10}$ monomers, $C_{10}H_{17}N_3O_x$, were expected to be formed from limonene via two steps of $NO_3$
oxidation to the double bonds and an addition of $NO_2$ to an $RO_2$ radical, leading to a peroxynitrate or
peroxyacylnitrate. $NO_2$ addition reactions may also contribute to the formation of 2N-$C_{10}$ monomers. The addition
of $NO_2$ to $RO_2$ radicals could occur either before (Scheme 2d) or after (Scheme 2c) the second $NO_3$ attack.

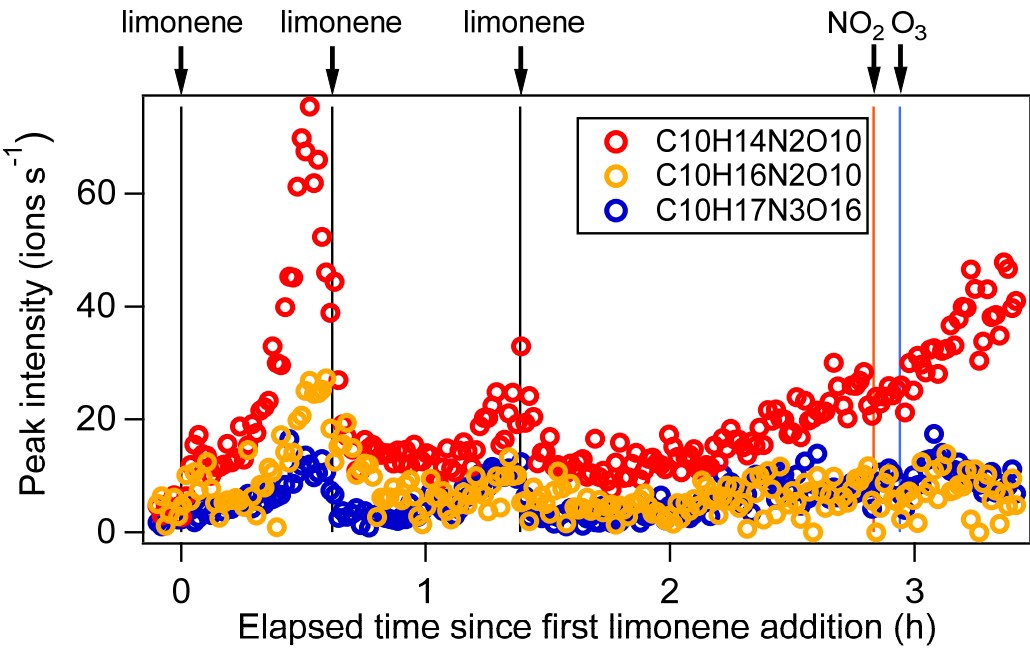


Figure 4. Time series of peak intensity of several monomers $C_{10}H_{14}N_2O_{10}$, $C_{10}H_{16}N_2O_{10}$ and $C_{10}H_{17}N_3O_{16}$ as the
representatives of multiple N monomers during the periods P1-P3.

(a) $C_{10}H_{15}NO_n$ $\xrightarrow[\text{H-shift, +O}_2]{\text{NO}_3{}^{\cdot},\ +O_2}$ $C_{10}H_{15}N_2O_{n+7}{}^{\cdot}$ $\xrightarrow{\text{termination}}$ $C_{10}H_{14}N_2O_{n+6}$ or $C_{10}H_{16}N_2O_{n+6/n+7}$

(b) $C_{10}H_{17}NO_n$ $\xrightarrow{\text{NO}_3{}^{\cdot},\ +O_2}$ $C_{10}H_{17}N_2O_{n+5}{}^{\cdot}$ $\xrightarrow{\text{termination}}$ $C_{10}H_{16}N_2O_{n+4}$ or $C_{10}H_{18}N_2O_{n+4/n+5}$

$\downarrow$ H-shift, +O$_2$

$C_{10}H_{17}N_2O_{n+7}{}^{\cdot}$ $\xrightarrow{\text{termination}}$ $C_{10}H_{16}N_2O_{n+6}$ or $C_{10}H_{18}N_2O_{n+6/n+7}$

(c) $C_{10}H_{17}NO_n$ $\xrightarrow[\text{H-shift, +O}_2]{\text{NO}_3{}^{\cdot},\ +O_2}$ $C_{10}H_{17}N_2O_{n+7}{}^{\cdot}$ $\xrightarrow{\text{NO}_2}$ $C_{10}H_{17}N_3O_{n+9}$

(d) $C_{10}H_{16}NO_n{}^{\cdot}$ $\xrightarrow{\text{NO}_2}$ $C_{10}H_{16}N_2O_{n+2}$ $\xrightarrow[\text{H-shift, +O}_2]{\text{NO}_3{}^{\cdot},\ +O_2}$ $C_{10}H_{16}N_3O_{n+9}{}^{\cdot}$ $\xrightarrow{\text{termination}}$ $C_{10}H_{15}N_3O_{n+8}$ or $C_{10}H_{17}N_3O_{n+8/n+9}$

Scheme 2. Possible formation pathways of $C_{10}$-monomers containing 2 nitrogen atoms (a, b) and 3 nitrogen atoms (c, d). Termination denotes reactions of RO$_2$• with other RO$_2$• or HO$_2$, or unimolecular reactions, leading to closed-shell products.

### 3.2.4 Formation pathways of $C_{10}$ monomers without N-atoms and monomers with less than 10 C-atoms

Besides $C_{10}$ products containing nitrogen atoms, HOM monomers without nitrogen atoms were also identified. Among these products, $C_{10}H_{14}O_x$ (x=7-12) were the most prevalent family, which were also detected in limonene ozonolysis (Jokinen et al., 2015). The $C_{10}H_{14}O_x$ family showed a time series typical of first-generation products (Fig. S8). $C_{10}H_{14}O_x$ and $C_{10}H_{16}O_x$ could be formed from limonene + NO$_3$ with $C_{10}H_{16}NO_x$• terminating their autoxidation by migration of the α-NO$_3$ H-atom, eliminating an NO$_2$ fragment (Scheme S2) (Novelli et al., 2021). Alternatively, these products could be formed via the reaction of O$_3$ with limonene (Scheme S2). Either way, $C_{10}H_{14}O_x$ and $C_{10}H_{16}O_x$ were formed via first-generation pathways.

We also observed monomers with carbon atom number less than 10. During the P1a period, $C_9$ monomer families were the most abundant contributors to C<10 HOM monomers, followed by $C_8$ families. The majority of $C_9$ monomers were $C_9H_{15}NO_x$ (x=7-13) (time series shown in Fig. S9) and $C_9H_{13}NO_x$ (x=8-14). The loss of one carbon atom may follow the mechanism shown in Scheme S3 (Fry et al., 2011; Bianchi et al., 2019). The major product family in $C_8$ monomers is $C_8H_{11}NO_x$ (x=6, 7, 9-13). While during period P1a $C_8H_{11}NO_x$ compounds could be hardly observed, their concentrations increased considerably in the later periods (Fig. S10). The gas-phase concentration of $C_8H_{11}NO_7$ was even the highest among all compounds in later periods (highest intensity signal in Fig. 2b). This is partly attributed to the relatively high volatility of $C_8$ compounds compared with $C_{10}$ HOM species and accretion products, which tend to condense on particles. The major family in $C_7$ monomers, $C_7H_9NO_x$ (x=6-13), showed a time series pattern similar to $C_8H_{11}NO_x$ compounds (Fig. S11). Such a time profile indicates that $C_7$ and $C_8$ products were likely a result of multi-generation gas-phase reactions.

### 3.3 Dimers and their formation

Among dimers, $C_{20}$ products were the most abundant, followed by $C_{19}$ products. Among $C_{20}$ and $C_{19}$ dimers, the most
prevalent families included $C_{20}H_{32}N_2O_x$ (x=9-20), $C_{20}H_{33}N_3O_x$ (x=12-20), $C_{20}H_{31}NO_x$ (x=10-15), $C_{20}H_{31}N_3O_x$
(x=14-20), $C_{20}H_{34}N_4O_x$ (x=15-20), and $C_{19}H_{30}N_2O_x$ (x=10-18) (Fig. S5). The O/C ratio of dimers did not exceed one,
while that of monomers was as high as two. This could be due to oxygen atom loss and participation of less
oxygenated $RO_2\bullet$ in the dimer formation as discussed below. Time series of dimers also showed different behavior
compared to monomers. For example, compounds of the $C_{20}H_{32}N_2O_x$ family only reached a considerable peak
intensity in period P1 and decreased rapidly thereafter, while the signal intensity in periods P2 to P6 were low (Fig.
5). Generally, other dimers showed similar patterns (Fig. S12-S14), though the difference of their concentration
between P2-P6 and P1 were not as large as for the $C_{20}H_{32}N_2O_x$ family. The time when signals of several dimers (e.g.
$C_{20}H_{32}N_2O_x$, $C_{20}H_{33}N_3O_x$, $C_{20}H_{34}N_4O_x$) dropped substantially matched the time of new particle formation (NPF) and
the onset of particle growth, indicating that some dimers were likely involved in the early growth of particles. Such
a behavior is expected since dimers have a much lower volatility than monomers. This observation is consistent with
the limonene + $NO_3$ laboratory study by Faxon et al. (2018) that found a significant fraction of HOM dimer derived
in the particle phase.

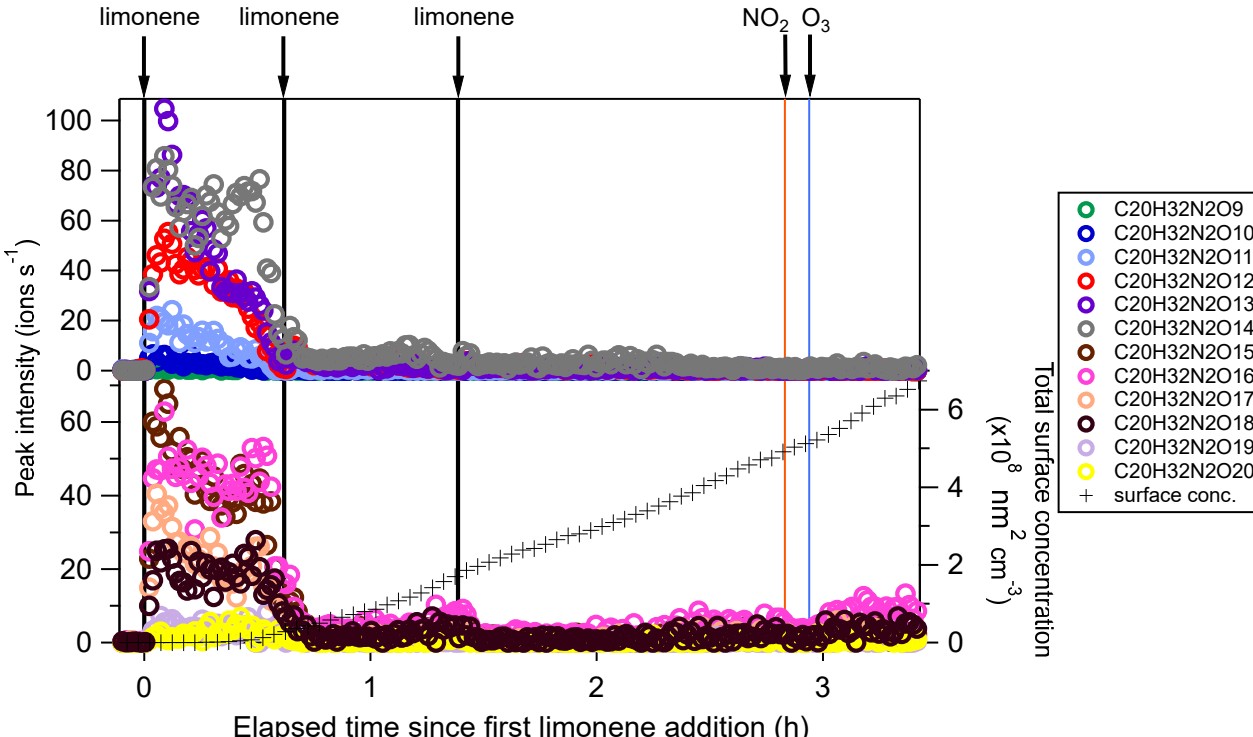

Figure 5. Time series of peak intensity of the $C_{20}H_{32}N_2O_x$ family compounds during the periods P1 to P3. The cross
markers (lower right y-axis) indicate total particle surface concentration.

In general, $C_{20}H_{32}N_2O_x$ showed an overlaying time profile of first- and second-generation products (Fig. 5).
$C_{20}H_{32}N_2O_x$ were likely formed via the accretion reaction between two monomer $RO_2\bullet$ ($C_{10}H_{16}NO_x\bullet$):
$$C_{10}H_{16}NO_{x1}\bullet + C_{10}H_{16}NO_{x2}\bullet \rightarrow C_{20}H_{32}N_2O_{x1+x2-2} + O_2 \quad\quad (R15)$$
Since $C_{10}H_{16}NO_x\bullet$ can be first- or second-generation products, the resulting dimers $C_{20}H_{32}N_2O_x$ can also be first- or
second-generation products. The time series show that $C_{20}H_{32}N_2O_x$ with lower O number presented more of a first-
generation product time profile (Fig. 5), while the relative contribution of second-generation formation was observed
to increase with oxygen number.
We compared the observed dimer formula with those expected based on accretion reactions of HOM-$RO_2\bullet$. X
in the $C_{20}H_{32}N_2O_x$ observed was $\geq 9$; however, according to the accretion mechanism and the observed $C_{10}H_{16}NO_x\bullet$
($x \geq 6$), x in $C_{20}H_{32}N_2O_x$ should be $\geq 10$ (6+6-2=10). Moreover, as the most abundant $RO_2\bullet$ within the $C_{10}H_{16}NO_x\bullet$
family was $C_{10}H_{16}NO_{10}\bullet$ (Table 1), the most abundant $C_{20}H_{32}N_2O_x$ would have an oxygen number of 18 if they were
exclusively formed by the accretion reaction of HOM $RO_2\bullet$. This contradicted the fact that the most abundant
molecule among the $C_{20}H_{32}N_2O_x$ family was $C_{20}H_{32}N_2O_{13}$. The findings above could only be explained by the
participation of less oxygenated $RO_2\bullet$ such as $C_{10}H_{16}NO_{5,6}\bullet$ in the accretion reaction (Berndt et al., 2018a; Berndt et
al., 2018b; Mcfiggans et al., 2019). $C_{10}H_{16}NO_5\bullet$ was not detected by our CI-APi-TOF, which is attributed to the lower
detection sensitivity of molecules with O number $\leq 5$ in the $NO_3^-$-CIMS (Riva et al., 2019). Still, $C_{10}H_{16}NO_5\bullet$ is the
first $RO_2$ radical formed in the limonene + $NO_3$ reaction (Scheme 1a) so a high mass flux has to pass through this
$RO_2\bullet$. If we assume that the abundance of $C_{10}H_{16}NO_5\bullet$ was high, and considering that the concentration of
$C_{10}H_{16}NO_{10}\bullet$ was the highest in the $C_{10}H_{16}NO_x\bullet$ family, their accretion reaction (R16) could form $C_{20}H_{32}N_2O_{13}$ and
support that $C_{20}H_{32}N_2O_{13}$ was the most abundant $C_{20}$ dimer product:
$$C_{10}H_{16}NO_5\bullet + C_{10}H_{16}NO_{10}\bullet \rightarrow C_{20}H_{32}N_2O_{13} + O_2 \quad\quad (R16)$$
Time series of dimers with different numbers of N atoms were different, indicating different formation pathways.
For example, the $C_{20}H_{31}NO_x$ family were mainly first-generation products (Fig. S12), which may be formed via the
following reaction:
$$C_{10}H_{16}NO_{x1}\bullet + C_{10}H_{15}O_{x2}\bullet \rightarrow C_{20}H_{31}NO_{x1+x2-2} + O_2 \quad\quad (R17)$$
$C_{10}H_{15}O_x\bullet$ were first-generation radicals (Sect. 3.2.4), while $C_{10}H_{16}NO_x\bullet$ were mainly second-generation radicals.
$C_{10}H_{16}NO_x\bullet$ could also be formed via first-generation pathway as discussed above (Scheme 1a), but that was not
borne out by the time profile, suggesting a fast termination of first-generation $C_{10}H_{16}NO_x\bullet$ radicals. Reaction R17
could be one of the termination pathways of first-generation $C_{10}H_{16}NO_x\bullet$ based on the first-generation time profile
of $C_{20}H_{31}NO_x$. In the study by Faxon et al. (2018), the formation of 1N-$C_{20}$ dimers was explained by a mechanism
involving two 1N-$RO_2$ radicals which produced $HNO_3$ as a by-product. However, $C_{10}$ $RO_2$ radicals without nitrogen
atoms were identified in our study, which provided a direct formation pathway of 1N-$C_{20}$ dimers through R17.
On the other hand, $C_{20}H_{33}N_3O_x$ and $C_{20}H_{34}N_4O_x$ were mainly second-generation products (Fig. S13, S14).
$C_{20}H_{33}N_3O_x$ and $C_{20}H_{34}N_4O_x$ were likely to be formed via $NO_3$ oxidation of dimers containing less nitrogen atoms,
and were thus second-generation products. The related radicals were also detected, such as $C_{20}H_{32}N_3O_x\bullet$ (x=16-19)
and $C_{20}H_{31}N_2O_x\bullet$ (x=13-16). Possible formation pathways of dominant oligomer families are displayed in Table 2.
We cannot exclude that the formation pathway of $C_{20}H_{33}NO_x$, $C_{20}H_{34}N_4O_x$ and $C_{19}H_{31}NO_x$ may also involve limonene

oxidation by OH• (Table 2), which can be formed in the ozonolysis of limonene as a minor pathway. In addition, the high abundance of $C_{20}H_{31}NO_x$ (x=10-15) among the dimers may be partly attributed to a contribution of the reaction of limonene with $O_3$.

The initial drop of the products (dimers and monomers) in Fig. 1, Fig. S8, and Fig. S12 during P1 (the characteristic time of the fastest decay was 15 min, 10 min, and 13 min, respectively) is attributed to the balance of their sources via the reaction of limonene with $NO_3$, their wall loss, and their potential loss by the reaction with $NO_3$. The characteristic time of the fastest decay of the HOM over the 2nd limonene addition in Fig. 1, 4, 5, S12, and S13 are 4-8 min. These decays can be explained by the wall loss rate (characteristic time ~8 min) and condensation sink of vapor loss to particles according to the study of Kulmala et al. (2012) (characteristic time ~13 min). The characteristic times of the fastest decay of the HOM at the end of P2 in Fig. S12 and S13 are 1.4-3.4 min, which can also be well explained by the updated wall loss rate and condensation sink of vapor loss to particles at the end of P2 (characteristic time ~1.4 min). In addition, for some HOM (e.g., $C_{10}H_{15}NO_{10}$ (Fig. 1), $C_{10}H_{14}N_2O_{10}$ (Fig. 4), $C_{20}H_{33}N_3O_{17}$ (Fig. S13), $C_{20}H_{34}N_4O_{17}$ (Fig. S14)), the times of limonene additions (except for the first time) matched the time when HOM signals dropped rapidly. This phenomenon implies a sudden decrease of the source of these HOM at limonene additions as sinks including the losses to walls and particles were largely invariant in such a short time. The decrease of the source may be attributed to the rapid depletion of $NO_3$ at limonene injections (Fig. 1). As many of these HOM are 2nd-generation products, i.e. formed via the reactions of 1st-generation products with $NO_3$, the depletion of $NO_3$ could lead to sudden decreases of the source of the 2nd-generation HOM, which accounted for most HOM in the study.

In summary, HOM dimers are likely to be formed via accretion reactions of monomer $RO_2$•, and some dimers can undergo secondary oxidation by $NO_3$. Some dimers were likely involved in the early growth of SOA particles.

Table 2. Major dimer and trimer families and their possible formation pathways.

| Dimer/Trimer family | Possible formation pathways |
| --- | --- |
| $C_{20}H_{32}N_2O_x$ | $C_{10}H_{16}NO_x• + C_{10}H_{16}NO_x•$ |
| $C_{20}H_{33}N_3O_x$ / $C_{20}H_{31}N_3O_x$ | $C_{20}H_{32}N_2O_x + NO_3 + HO_2•/RO_2•$ |
| $C_{20}H_{31}NO_x$ | $C_{10}H_{16}NO_x• + C_{10}H_{15}O_x•$ |
| $C_{20}H_{33}NO_x$ | $C_{10}H_{16} + OH• + C_{10}H_{16}NO_x•$ |
| $C_{20}H_{34}N_4O_x$ | $(C_{10}H_{16}N_2O_x + OH•) + (C_{10}H_{16}N_2O_x + OH•)$ |
| $C_{19}H_{30}N_2O_x$ | $C_{10}H_{16}NO_x• + C_9H_{14}NO_x•$ |
| $C_{19}H_{31}N_3O_x$ | $C_{19}H_{30}N_2O_x + NO_3 + HO_2•/RO_2•$ |
| $C_{19}H_{29}NO_x$ | $C_9H_{14}NO_x• + C_{10}H_{15}O_x•$ |
| $C_{19}H_{31}NO_x$ | $C_{10}H_{16} + OH• + C_9H_{14}NO_x•$ |
| $C_{30}H_{48}N_4O_x$ | $C_{20}H_{32}N_3O_x• + C_{10}H_{16}NO_x•$ |
| $C_{30}H_{47}N_3O_x$ | $C_{20}H_{31}N_2O_x• + C_{10}H_{16}NO_x•$ |

### 3.4 Trimers and their formation

Trimers ($C_{26-30}$) were dominated by $C_{30}$ compounds (Fig. S6, Table S1). To the best of our knowledge, this is the first study that identified gas-phase trimers in the limonene + NO$_3$ reaction. The O/C ratio of trimers were lower than that of monomers and dimers, suggesting possible multiple accretion reactions in their formation pathways, which lose 2 oxygen atoms in each reaction. As each accretion reaction terminates the peroxy radical chain, the observation of trimers also implies that some dimers could further react with NO$_3$, creating dimer RO$_2$•. The most prevalent product families were $C_{30}H_{48}N_4O_x$ (x=16-24) and $C_{30}H_{47}N_3O_x$ (x=18,19,21,23,24), which were likely formed via the most abundant monomer RO$_2$ radicals - $C_{10}H_{16}NO_x$• and the most abundant dimer RO$_2$ radicals - $C_{20}H_{32}N_3O_x$• and $C_{20}H_{31}N_2O_x$•. Trimers from other monoterpenes + NO$_3$ have been observed in previous laboratory studies. For example, $C_{30}H_{48}N_4O_{16}$ and $C_{30}H_{47}N_3O_{16}$ were observed in the mass spectra of α-pinene + NO$_3$ SOA by Wu et al. (2021a), and $C_{30}H_{47}N_3O_{13}$ was identified in β-pinene + NO$_3$ SOA by Claflin and Ziemann (2018).

Similar to their precursors $C_{20}H_{32}N_2O_x$, $C_{30}H_{48}N_4O_x$ showed negligible signal except in period P1, and presented an overlaying time profile of first- and second-generation product pattern (Fig. 6). For comparison, gas-phase trimer products were not observed in the β-pinene + NO$_3$ reaction (Shen et al., 2021), and the trimers observed in SOA from β-pinene + NO$_3$ are likely formed via particle phase reactions (Claflin and Ziemann, 2018). An efficient gas-phase trimer production via subsequent accretion reactions between peroxy radicals requires that the precursor dimer has a high enough reactivity to create a dimer RO$_2$•, e.g. via NO$_3$ reaction to a double bond. This suggests that the VOC containing at least two double bonds are likely more favorable to form trimers, which is consistent with our previous findings that trimers were formed in the NO$_3$ reaction with isoprene which also contains two double bonds (Zhao et al., 2021) while they were not observed in the reaction of NO$_3$ with β-pinene which contains only one double bond (Shen et al., 2021).

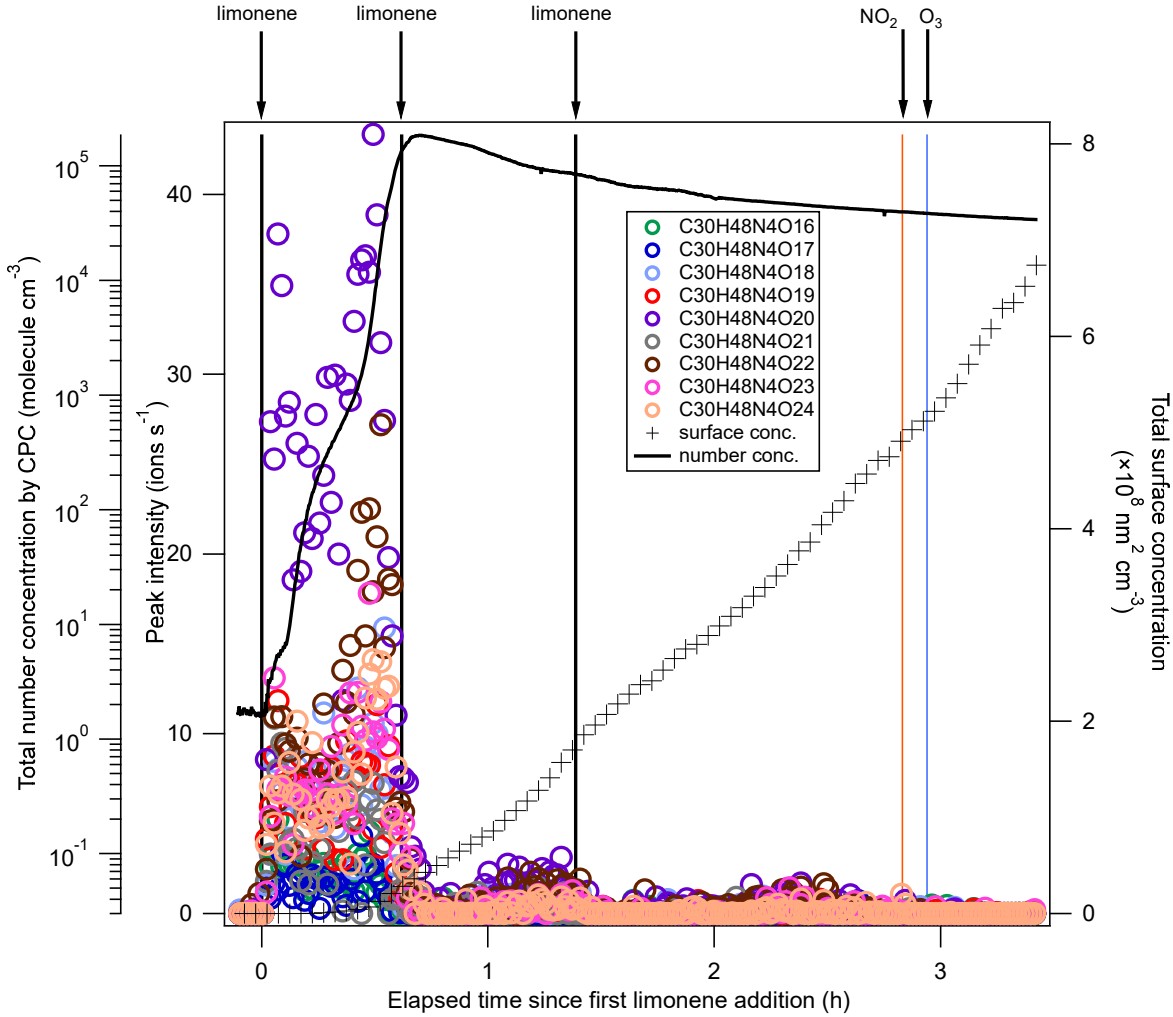


Figure 6. Time series of peak intensity of the $C_{30}H_{48}N_4O_x$ family compounds during the periods P1 to P3. The solid
black line refers to total number concentrations detected by CPC. The cross markers (right y-axis) indicate total
particle surface concentration.

## 3.5 "Primary" incremental HOM yields

We chose period P1 for the calculation of HOM yields in order to minimize the influence of the condensational sink
on HOM concentration. However, both first-generation and second-generation products existed in this period, as
discussed in Sect. 3.2 through 3.4 and supported by the time-behavior of the total HOM concentration (Fig. S15).
Period P1 can be roughly divided into three phases based on the trends in HOM concentration. Shortly after the
limonene injection, large quantities of HOM were produced (first-production phase) followed by a balanced
intermediate phase when HOM concentrations stopped increasing. After the intermediate phase, HOM concentrations
began to increase again (second-production phase). The first-production phase overlapped with the time span where
limonene, $NO_3$ and $N_2O_5$ concentrations decreased, implying the dominance of first-generation HOM production
process. During the second production period, wall loss was compensated by second-generation HOM formation,
leading to another rise of the total HOM concentrations. Therefore, we use the first-production phase to estimate
primary HOM production, determined over the first 3 min of the experiment. The calculated "primary" HOM molar
yield is 1.5 %$^{+1.7\ \%}_{-0.7\ \%}$. This value is significantly lower than the HOM yield of 5 to 17 % in earlier limonene ozonolysis
experiments (Ehn et al., 2014; Jokinen et al., 2015; Pagonis et al., 2019). It should be emphasized that second-
generation HOM, which contributed greatly to the limonene + NO₃ reaction system, is not included in this primary
HOM yield.

**3.6 Contribution of HOM to particle formation and growth**

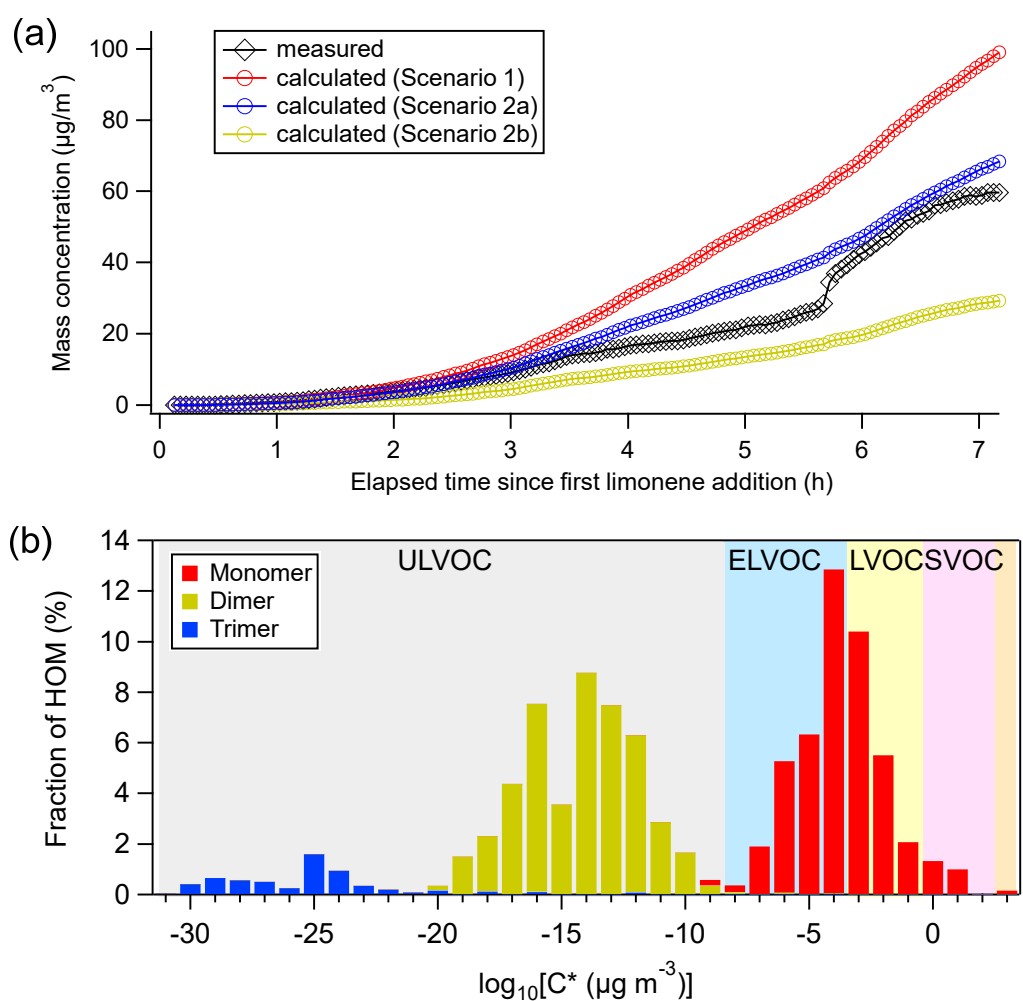


Figure 7. (a) Comparison of measured particle concentrations (black) against those predicted from condensation of
measured HOM on aerosol particles, where red markers were calculated under Scenario 1, blue and yellow markers
were calculated under Scenario 2 (only considering the condensation of ULVOC, ELVOC and LVOC), with the
volatility calculated using the method by Mohr et al. (2019) (Scenario 2a) and Peräkylä et al. (2020) (Scenario 2b)
respectively. (b) HOM volatility distribution using formula provided by Mohr et al. (2019). Average concentrations
of HOM in the P1 period were used to calculate the fraction of HOM.

We observed nucleation and growth of SOA particles in the limonene + NO₃ reaction. We calculated the contribution

of HOM to SOA formation and particle growth and compared it to the measured particle growth (Fig. 7a). We assumed different scenarios of HOM uptake on aerosol particles, using the calculation methods described in the literature (Ehn et al., 2014; Seinfeld and Pandis, 2006; Nieminen et al., 2010). The assumption that all HOM irreversibly condense on the particles (Scenario 1) resulted in a strong overestimation of particle mass growth (red markers in Fig. 7a). Applying the parameterizations of Mohr et al. (2019) (Scenario 2a) or Peräkylä et al. (2020) (Scenario 2b) for classification and accounting only LVOC- and ULVOC/ELVOC-HOM for irreversible uptake framed the observed values (blue and yellow markers in Fig. 7a). While Scenario 2a agreed quite well with the observations and only slightly overestimated SOA concentrations after 7 h by +11%, Scenario 2b underestimated the SOA concentration at the end by -53%. The agreement between the modeled and observed SOA concentration suggests that HOM, and particularly LVOC- and ULVOC/ELVOC-HOM play a major role in growth of SOA particles in this study. This is consistent with the work by Faxon et al. (2018) who found that many of the dimers are ELVOC, which is also supported by our calculation result based on the method of Mohr et al. (2019).

Since neither $SO_2$ nor $H_2SO_4$ was added in our experiment, new particle formation (NPF) could be attributed to the nucleation initiated by HOM of low volatility. HOM trimers with as many as 30 carbon atoms were identified in the early stage of this study, and their sudden loss matched the onset of rapid formation of SOA. Trimers identified in our experiment are classified as ULVOC/ELVOC, with much lower volatility than monomers and dimers (Fig. 7b). Therefore, NPF in the current study can more likely be attributed to HOM trimers since they have the strongest potential of initiating nucleation, although we cannot rule out some contributions of dimers in the NPF. In contrast, in an earlier experiment investigating the $NO_3$-initiated oxidation of β-pinene also conducted in the SAPHIR chamber under similar conditions, new particles were barely formed (<20 cm$^{-3}$) (Shen et al., 2021). As already mentioned above, no trimer HOM products were observed in that study, and only molecules with C≤20 were detected (Sect. 3.4). Extremely low volatile organic vapors formed in α-pinene ozonolysis have been shown to induce nucleation and drive initial particle growth in the atmosphere (Tröstl et al., 2016; Kirkby et al., 2016). Since our experiment of $NO_3$ oxidation of limonene was performed under near atmospheric conditions, such NPF events induced by the oxidation of limonene by $NO_3$ could also occur in the ambient atmosphere. Although monoterpene concentrations in this study (0-0.92 ppbv) are higher than in most ambient regions, they are still in the range of ambient concentrations (~0.01-1 ppbv) (e.g. Coggon et al., 2021; Wang et al., 2022), especially for forested regions (e.g. Xu et al., 2015; Kontkanen et al., 2016; Janson, 1992). Assuming that dimers react with $NO_3$ at a rate similar to limonene, and that they have a condensation sink similar to $H_2SO_4$ ($10^{-3}$-$10^{-1}$ s$^{-1}$) (Dada et al., 2020), the lifetime with respect to $NO_3$ at an $NO_3$ concentration of 5-300 ppt and to condensation on particles are ~0.1-10 min and ~0.1-20 min, respectively. Therefore, although aerosols may scavenge HOM dimers in the ambient atmosphere, dimers can still react with $NO_3$ at nighttime, forming trimers. Such reactions are particularly important when the ambient aerosol concentration is low. Several field observations have shown NPF events taking place at nighttime where biogenic emissions dominate (Kammer et al., 2018; Huang et al., 2019). The work by Ortega et al. (2012) demonstrated an important role of monoterpene ozonolysis products in nocturnal NPF events in chamber experiments. In a previous laboratory study, limonene +

$NO_3$ appears more effective at initiating nucleation than the limonene + $O_3$ reaction (Fry et al., 2014), which supports
that limonene + $NO_3$ can play a significant role in nighttime nucleation. Our study suggests that $NO_3$ oxidation of
limonene could contribute to the nighttime NPF via HOM trimer formation. In contrast, we infer that $NO_3$ reactions
with other monoterpenes containing only one double bond such as α-pinene and β-pinene are less likely candidates
for nighttime NPF, because gas-phase trimers are not observed.

**4 Conclusion and implications**
HOM formation in the reaction of limonene with $NO_3$ was investigated in the SAPHIR chamber. About 280 gas-
phase HOM products were identified, including monomers ($C_{6-10}$, $O_{6-16}$, $N_{0-3}$), dimers ($C_{17-20}$, $O_{7-20}$, $N_{0-4}$) and trimers
($C_{27-30}$, $O_{16-25}$, $N_{1-6}$). Nitrogen-containing products dominated the HOM, with compounds of the $C_{10}H_{15-17}NO_{6-14}$
series being the most prevalent. Dimers contributed 47 % in the early stage of the experiment when particle surface
concentration was rather low ($< 6 \times 10^4$ $nm^2$ $cm^{-3}$), which was similar to monomers (47 %). Tentative formation
pathways of major families were proposed in this work based on their time-dependent concentration profiles.
In HOM monomers, the abundance of carbonyl compounds significantly exceeded that of hydroxy or
hydroperoxy compounds, indicating the significance of unimolecular termination of HOM-$RO_2$ radicals. Both $RO_2\bullet$
autoxidation and alkoxy-peroxy pathways were found to be important in the formation of HOM monomers.
Monomers with 1 nitrogen atom (1N-monomers) contained both first- and second-generation products, which could
be formed via $NO_3$ oxidation of limonene and its first-generation products with the latter being more important.
Monomers with 2 nitrogen atoms were classified as second-generation products, which could be formed via $NO_3$
oxidation of the remaining C=C double bond of 1N-monomers.
Dimers showed both first- and second-generation time pattern. Dimers were mostly formed via accretion
reactions between monomer $RO_2$ radicals, resulting in a decrease in O/C ratio compared to monomers. The initial
less oxygenated $RO_2\bullet$, including the $C_{10}H_{16}NO_5\bullet$ radical that cannot be observed in our instrument, likely played an
important role in dimer formation based on the comparison of measured dimers against expected dimer identity and
concentrations according to accretion monomer $RO_2\bullet$ reactions. Trimers were likely formed via accretion reactions
between monomer $RO_2$ and dimer $RO_2$ radicals formed from secondary reactions of dimers with $NO_3$. Trimer
formation is thus linked to the presence of two double bonds in limonene, of which the first reacts with $NO_3$ leading
to dimer products while the remaining C=C double bond provides a reactive site for further oxidation of the dimers
by $NO_3$, forming dimer $RO_2$ radicals.
A "primary" HOM molar yield of 1.5 $\%^{+1.7\,\%}_{-0.7\,\%}$ in the limonene + $NO_3$ reaction was estimated, including only the
first-generation HOM. Second-generation HOM contributed greatly to monomers, dimers and trimers, and hence the
HOM yield we obtained is a lower limit of the total HOM yield, and is likewise much lower than the total HOM yield
in the reaction of limonene with ozone (5 to 17 %) (Ehn et al., 2014; Jokinen et al., 2015; Pagonis et al., 2019).
NPF observed in this work was likely related to the trimer formation due to much lower volatility of trimers
compared to monomers and dimers. The SOA concentration in the limonene + $NO_3$ reaction could be explained by

the condensation of the HOM belonging to LVOC and ULVOC/ELVOC classes assuming irreversible uptake, indicating an important role of HOM for growth of SOA particles in this reaction system. To our knowledge, this work is the first identifying trimer products from the limonene + $NO_3$ reaction system, suggesting that limonene + $NO_3$ is a possible crucial source of new particles formed in nighttime biogenic emission-dominated areas (Kammer et al., 2018; Huang et al., 2019). Our work highlights the need to consider the role of limonene + $NO_3$ in NPF in models simulating nighttime aerosols formation in biogenic-emission dominated areas, especially with large limonene emissions. In addition, comparison with the reactions of $NO_3$ with isoprene (Zhao et al., 2021) and other monoterpenes (Shen et al., 2021) reveals a strong dependence of HOM products on the molecular structure of the VOC species in $NO_3$-initiated chemistry.

The concentration of limonene and $NO_3$ in this study were on the order of few ppb and a few to one hundred ppt, respectively, which are similar to the ambient levels in rural and forest regions affected by anthropogenic emissions (Brown and Stutz, 2012). The chemical lifetime of $RO_2\bullet$ was of the order of 50 to 500 s, which is also similar to ambient conditions at nighttime (Fry et al., 2018). The $RO_2\bullet$ loss pathway in our study was dominated by the reactions $RO_2\bullet + NO_3$ and $RO_2\bullet + RO_2\bullet$, which is relevant for the $RO_2\bullet$ fate in urban areas and forested areas influenced by an urban plume at nighttime. However, in more pristine forested regions, the $RO_2\bullet$ fate is mostly determined by $RO_2\bullet + HO_2$ and $RO_2\bullet + RO_2\bullet$, as shown by Bates et al. (2022) for the example of a Southeast US forest. As $NO_3$ concentration is generally enhanced with increased anthropogenic emissions, $RO_2\bullet + NO_3$ will become more important going from remote to urban areas. Therefore, the HOM products and their formation process in our study are relevant for rural and forested regions influenced by anthropogenic plumes and ambient urban regions with high volatile commercial products emissions as limonene is a typical component of volatile chemical products (VCP) (Nazaroff and Weschler, 2004). In these regions, HOM from monoterpene + $NO_3$ reactions can be major components of nighttime SOA. As nitrooxy-$RO_2$ fate can strongly affect the oxidation product distribution and SOA yield as shown for the reaction of α-pinene with $NO_3$ (Bates et al., 2022), more studies of HOM formation by $NO_3$ at various $RO_2\bullet$ fates are needed to be representative of various environment including (remote) forested regions.

This study also highlights the important role of second-generation chemistry in HOM formation, which needs to be further investigated and should be included in chemical mechanism used in numerical models. Additional work is also needed to investigate the role of different HOM formed via $NO_3$-initiated BVOC oxidation reactions in NPF and growth of SOA particles in order to better constrain the climatic and environmental effects of BVOC + $NO_3$ chemistry.

Acknowledgement

Y. Guo, H. Shen, H. Luo, and D. Zhao would like to thank the funding support of Science and Technology Commission of Shanghai Municipality (No. 20230711400), National Natural Science Foundation of China (No. 41875145), and Shanghai International Science and Technology Partnership Project (No. 21230780200). Sungah Kang, Astrid Kiendler-Scharr, and Thomas F. Mentel acknowledge the support by the EU Project FORCeS (grant

agreement no. 821205).

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
