# Peer review of "Identification of highly oxygenated organic molecules and their role in"

_Atmospheric Chemistry and Physics, 2022_

## Author Comment (AC1)

We thank the reviewers for the helpful comments on our manuscript. The comments are greatly appreciated. We addressed all the comments and believe that the revisions based on the comments helped improve the quality of our manuscript. Below please find our responses to the comments one by one and the corresponding revisions made to the manuscript. The original comments are shown in italics. The revised parts of the manuscript are highlighted.

*General comments*

●*The manuscript by Guo et al present interesting results of highly oxygenated organic molecules (HOM) from the limonene+NO3 system. This system is of large atmospheric relevance, yet such measurements have hardly been performed before. The experiments are of high quality, and I think the manuscript nicely fits within the scope of ACP. I have several questions and comments to the authors, as outlined below, with my major concern being the interpretation of some of the results.*

**Response:**

We thank the reviewer for the kind remarks.

*Specific comments*

●*Lines 110-127: Different instruments have been shown to have different sensitivities to molecule groups, and even CIMS instruments with different reagent ions observing the same elemental composition can often show totally different temporal behavior. Considering this, the different reported HOMs in these papers might not be the same thing. It might be useful to group the papers in such a way that a reader knows which instrument measured the reported HOMs. In addition, I think Yan et al. (2016) (https://doi.org/10.5194/acp-16-12715-2016) could also be mentioned when discussing reported NO3-induced HOM.*

**Response:**

Accepted. This part of literature review is in the order of from field campaigns to laboratory studies. In the revised manuscript, the instruments used to detect HOM in each work have been added as follows:

"In the SOAS campaign, HOM-ONs (organic nitrates) were identified in both gas and particle phase ==using a $NO_3^-$-Chemical Ionization time-of-flight Mass Spectrometer (CI-APi-TOF) and a High Resolution Time-of-Flight Chemical Ionization Mass Spectrometer (HR-ToF-CIMS) coupled to a Filter Inlet for Gases and AEROsols (FIGAERO)==. Species with the sum formula $C_{10}H_{15,17,19}NO_{4-11}$ were observed which are formed through the oxidation of monoterpenes by $NO_3$ (Lee et al., 2016; Massoli et al., 2018). … Boyd et al. (2015) observed $C_{10}H_{17}NO_{4/5}$ and $C_{10}H_{15}NO_{5/6}$ in the gas phase in β-pinene + $NO_3$ experiments ==using a quadrupole chemical ionization mass spectrometer with I⁻ as the reagent ion (I⁻-CIMS)==. They proposed possible

formation schemes of these ONs. Nah et al. (2016) further detected 5 and 41 HOM-ONs in the $NO_3$ oxidation of α-pinene and β-pinene, respectively, such as $C_{10}H_{15/17/19}NO_{4-9}$ in the gas- and particle-phase ==using I⁻-FIGAERO HR-ToF-CIMS==. Claflin and Ziemann (2018) provided formation mechanisms for HOM-ONs via gas-phase and particle-phase reactions in the β-pinene + $NO_3$ reaction system, ==where particle-phase products were analyzed using reversed-phase high-performance liquid chromatography equipped with a UV−vis photodiode array detector (HPLC-UV), Electron-Ionization Thermal Desorption Particle Beam Mass Spectrometer (EI-TDPBMS), Chemical Ionization Finnigan PolarisQ Ion Trap Mass Spectrometer (CI-ITMS), and Electrospray-Ionization Mass Spectrometer (ESI-MS)==. Recently, Shen et al. (2021) found a large number of HOM (>150 species) in the β-pinene + $NO_3$ reaction ==using $NO_3^-$-CI-APi-TOF==. ==HOM formed in the reaction of four monoterpenes (α-Pinene, β-pinene, Δ-3-carene, and α-thujene) with $NO_3$ were also detected using $NO_3^-$-CI-APi-TOF by Dam et al. (2022)==. Bell et al. (2021) found that dimer dinitrates ($C_{20}H_{32}N_2O_{8-13}$) contribute a large portion to SOA from α-pinene + $NO_3$ and also detected monomer ON such as $C_{10}H_{15}NO_{5-10}$ and $C_{10}H_{14,16}N_2O_{7-11}$) ==using FIGAERO-CIMS and an Extractive ElectroSpray Ionization time-of-flight mass spectrometer (EESI-ToF-MS)==. The detailed speciation depends on analytical method to some extent, though. Moreover, the HOM composition in the particle-phase was found to depend on aging time and reaction conditions such as dark versus light (Bell et al., 2021; Wu et al., 2021). ……Regarding the HOM formation in the reaction of limonene with $NO_3$, Faxon et al. (2018) reported a series of HOM in the particle phase, including $C_{7-10}$ monomers with 3-11 oxygen atoms and $C_{11-20}$ dimers with 5-19 oxygen atoms ==using I⁻-FIGAERO HR-ToF-CIMS==."

The paper by Yan et al. (2016) has been also added to the manuscript. The revised text reads as follows:

=="In a campaign in a boreal forest in Hyytiälä, measurement using a $NO_3^-$-CI-APi-TOF and positive matrix factor (PMF) analysis showed a nighttime factor of HOM-ON formed via $NO_3$ oxidation of monoterpenes (Yan et al., 2016)."==

●*Fig. 1: Please multiply the NO3 time trace by (at least) 10 to make the values legible.*

**Response:**

Accepted. In the revised manuscript, Fig. 1a has been modified in which $NO_3$ concentrations are multiplied by 10 (also shown below), and show similar pattern as $N_2O_5$:

[Figure]

●*Line 162: Do you expect that isoprene would be left on the walls? Is it possible that SVOC isoprene oxidation products are coming off the walls instead?*

**Response:**

We do not expect that isoprene is left on the walls because its volatility is very high. Instead, we indeed believe that some peaks we identified are likely SVOC isoprene oxidation products left on the walls, such as $C_5H_{10}N_2O_8$ mentioned in our manuscript. In the revised manuscript, we have further clarified this point to avoid potential misunderstanding as follows:

"We observed several peaks which were obviously products from the isoprene + $NO_3$ reaction, such as $C_5H_{10}N_2O_8 \cdot {}^{15}NO_3^-$ at m/z 289. Such peaks were present before the limonene oxidation reaction started, suggesting that these compounds preexisted in the chamber. These isoprene oxidation products were likely formed in an isoprene + $NO_3$ experiment performed two days before (Zhao et al., 2021) and released slowly from chamber walls due to their semi-volatile character. "

●*When discussing the atmospheric relevance of this work, I think the authors should reference their conditions to the findings by Bates et al. (2021) (https://doi.org/10.5194/acp-22-1467-2022).*

**Response:**

Accepted. We have revised the discussion regarding the ambient relevance and referred to the finding of Bates et al. (2022) as follows.

"The $RO_2\bullet$ loss pathway in our study was dominated by the reactions $RO_2\bullet + NO_3$ and $RO_2\bullet + RO_2\bullet$, which is relevant for the $RO_2\bullet$ fate in urban areas and forested areas influenced by an urban plume at nighttime. However, in more pristine forested regions, the $RO_2\bullet$ fate is mostly determined by $RO_2\bullet + HO_2$ and $RO_2\bullet + RO_2\bullet$, as shown by Bates et al. (2022) for the example of a Southeast US forest. As $NO_3$

concentration is generally enhanced with increased anthropogenic emissions, $RO_2\bullet + NO_3$ will become more important going from remote to urban areas. Therefore, the HOM products and their formation process in our study are relevant for rural and forested regions influenced by anthropogenic plumes and ambient urban regions with high volatile commercial products emissions as limonene is a typical component of volatile chemical products (VCP) (Nazaroff and Weschler, 2004). In these regions, HOM from monoterpene + $NO_3$ reactions can be major components of nighttime SOA. As nitrooxy-$RO_2$ fate can strongly affect the oxidation product distribution and SOA yield as shown for the reaction of α-pinene and $NO_3$ (Bates et al., 2022), more studies of HOM formation by $NO_3$ at various $RO_2\bullet$ fates are needed to be representative of various environment including (remote) forested regions.''

We have also cited Bates et al. (2022) in the introduction section, as well as several other places.

●*I find that including some high resolution fits is very helpful for a reader to assess the certainty of peak assignments, and therefore suggest that some examples be included. Ranging from ions that are unambiguous to ions that are on the limit of what you included in the paper. For example, the conclusions mention trimers with six N-atoms, and the quality of these fits would be interesting to see.*

**Response:**

Accepted. We have added two examples of high resolution peak fitting, representing unambiguous ions and ion on the limit of the resolution, respectively, in the supplement of the revised manuscript (Fig. S14), which are also shown below:

[Figure]

Figure S14. Examples of high resolution peak fitting of HOM containing 1 N atom (left panel) and 5 N atoms (right panel). Red lines are the mass spectrum, blue and orange lines show the sum of the isotopic contributions and the fitted peaks, and the residuals, respectively. The black vertical lines denote the exact mass of the fitted peaks.

●*It is many times stated that "carbonyl products outnumbered hydroxyl products, indicating the importance of the unimolecular RO2 termination pathway". This may be true, but I would like to see some estimate of the importance of RO2+RO2->RO+RO followed by RO+O2->R=O+HO2. I also don't know why the RO-forming reactions are not included in the list of reactions startign on line 96. It is after all typically the dominant pathway.*

**Response:**

We agree that the pathway $RO_2\bullet+RO_2\bullet\rightarrow RO\bullet+RO\bullet$ followed by $RO\bullet+O_2\rightarrow R=O+HO_2$ may have some limited importance. We have added this pathway in our manuscript. However, the fragmentation and isomerization of $RO\bullet$ is also important especially for large $RO\bullet$ radicals with multiple functional groups. In the revised manuscript, we have added the following discussion.

"Among 1N-C10 monomers, concentrations of carbonyl compounds were much higher than the sum of hydroxy- and hydroperoxy-substituted compounds (Table 1). … This finding is likely attributed to unimolecular termination reactions of $RO_2\bullet$, although reaction paths via $RO\bullet$ also cannot be excluded. Smaller unbranched $RO\bullet$ tend to react with $O_2$ forming carbonyl compounds while for larger or branched $RO\bullet$, isomerization can also form carbonyl compounds and is a more energetically favorable and thus faster pathway compared with the reaction with $O_2$ (Ziemann and Atkinson, 2012)."

In our original manuscript, we only listed the reactions of $RO_2\bullet$ forming close-shell product. We have also added the reaction of $RO_2\bullet$ to $RO\bullet$ in the revised manuscript.

"The bimolecular reactions of $HOM-RO_2\bullet$ with $RO_2\bullet$, $HO_2\bullet$ and NO lead to highly oxidized closed shell products including carbonyls, hydroperoxides, alcohols, or organic nitrates as termination groups (R1 to R3), or form accretion products (R4) (Ehn et al., 2014; Mentel et al., 2015). Unimolecular termination reactions of $HOM-RO_2\bullet$ lead to carbonyls or epoxides (R5 to R6) (Crounse et al., 2013). On the other hand, reactions of $HOM-RO_2\bullet$ with NO, $RO_2\bullet$, $NO_3$ at nighttime can lead to alkoxy radicals as chain propagating steps (R7 to R9):"

$$\text{``}RO_2\bullet + NO \rightarrow RO\bullet + NO_2 \qquad\qquad (R7)$$
$$RO_2\bullet + RO_2\bullet \rightarrow RO\bullet + RO\bullet \qquad\qquad (R8)$$
$$RO_2\bullet + NO_3 \rightarrow RO\bullet + NO_2 + O_2 \qquad\qquad (R9)\text{''}$$

●*Lines 189-190: " a large fraction of limonene was already reacting away during the VOC injection before it was homogeneously mixed in the chamber." It took me a while to understand what this meant. But it then also raises the question that if all the HOM yields are determined from the first 3 minutes after injection, what is the influence of incomplete mixing? This will impact both the limonene + NO3 reactions, the RO2 fates, and the amount of HOM measured, if there are "hotspots" in the chamber with clearly higher*

*concentrations. Was there a fan used for mixing?*

**Response:**

HOM yield was calculated using the first 3 min of the experiment, because during that period particle concentration was low and $1^{st}$-generation reactions dominated. A fan was used for mixing during the whole experiment, and time for complete mixing is ~1 min (Fuchs et al., 2013), which was further confirmed based on measured VOC concentrations. Therefore at 3 min the core of the chamber is already quite well mixed and the measured VOC consumed can be used to determine HOM yield.

In the revised manuscript, we add one sentence of regarding the chamber running.

"A fan was used for active mixing in the chamber, leading to a typical mixing time of ~ 1 min (Fuchs et al., 2013)."

●*Related to the above, I find the reported wall loss rate of HOM (6e-4 s^-1) to be extremely low when comparing to any other chambers where HOM loss rates were reported, even when considering the large volume of SAPHIR. A lifetime of almost 30 min for low-volatile species seems very surprising, especially if the chamber was actively mixed (and if it was not, then I would expect very large inhomogeneities). The long lifetime seems to be based solely on tracking one molecule, C10H15NO8, by Zhao et al (2018). At the same time, the paper by Peräkylä et al. (2020) suggests that this specific molecule (at mass 339 Th) hardly condensed in their seed addition experiments, suggesting that it may be a bad surrogate for LVOC.*

**Response:**

We thank the reviewer's comment, which made us realize a mistake in the wall loss rate originally used. The wall loss rate of $6\times10^{-4}$ $s^{-1}$ was determined for the condition when the chamber was not actively mixed by using the decay of $C_{10}H_{15}NO_8$ in the dark chamber in the work by Zhao et al. (2018). When the chamber was actively mixed, the wall loss is determined to be $(2.2\pm0.2)\times10^{-3}$ $s^{-1}$. We apologize for this mistake.

Besides $C_{10}H_{15}NO_8$, we also examined the decay of other compounds, such as $C_{10}H_{15}NO_{9-12}$ (volatility in the LVOC/ELVOC range) and non-nitrated compounds such as $C_{10}H_{14}O_{8-11}$, which all showed similar decay rates as shown below. We think that $C_{10}H_{15}NO_8$ can be a suitable surrogate to evaluate wall loss of HOM because in our previous study in the photooxidation of β-pinene, we found that $C_{10}H_{15}NO_8$ condensed on the particles with a high uptake coefficient of ~0.3. The difference between our study and that by Peräkylä et al. (2020) may be attributed to different particle surface concentration, which is beyond the scope of this study.

The lower wall loss rate in SAPHIR chamber during active mixing is comparable to other chambers

such as those reported by Peräkylä et al. (2020) (lifetime ~400 s) and Krechmer et al. (2016) (lifetime ~7-13 min). We would like to note that the SAPHIR chamber is much larger than other chambers (270 m³ vs a few m³) and also the chamber is running in batch mode instead of continuous flow mode used for many other chambers for HOM studies such as our JPAC (Ehn et al., 2014; Pullinen et al., 2020) or COALA chamber (Peräkylä et al., 2020). The large volume and batch running mode may result in a thicker boundary layer of the chamber wall, which delays vapor wall deposition.

[Figure]

Figure S15. Decay of HOM $C_{10}H_{14}O_{8-11}$ (a) and $C_{10}H_{15}NO_{9-12}$ (b) due to wall loss during active mixing in SAPHIR chamber. The lifetimes (tau) of wall loss of each species are shown.

In the revised manuscript, we have updated the wall loss rate and added more description of the wall loss rate in the supplement.

"The concentrations of HOM were corrected for chamber wall losses, which were determined for a number of HOM similar to our previous study (Zhao et al., 2018), with details described in the supplement.

As the HOM yield was determined during the first 3 min of the experiment, we considered the wall loss rate to be constant $(2.2\times10^{-3}\ s^{-1})$ during this period."

"When the chamber is actively mixed, the wall loss is determined to be $(2.2\pm0.2)\times10^{-3}\ s^{-1}$."

"We examined the decay of nitrated compounds such as $C_{10}H_{15}NO_{9-12}$ (volatility in the LVOC/ELVOC range) and non-nitrated compounds such as $C_{10}H_{14}O_{8-11}$ in the reaction of limonene by OH in the presence of NO, which all showed similar decay rates as shown below (Fig. S15).

The wall loss rate in SAPHIR chamber during active mixing is comparable to the chambers such as those reported by Peräkylä et al. (2020) (lifetime ~400 s) and Krechmer et al. (2016) (lifetime ~7 to 13 min). We would like to note that the SAPHIR chamber is much larger than other chamber (270 m$^3$ vs a few m$^3$) and also the chamber is running in batch mode instead of continuous flow mode used for many other chambers for HOM studies such as the COALA chamber (Peräkylä et al., 2020) or our JPAC chamber (Ehn et al., 2014; Pullinen et al., 2020). The large volume and batch running mode may result in a thicker boundary layer of the chamber wall, which delays vapor wall deposition."

●*This also brings me to my main concern with this manuscript, namely the interpretation of the observed HOM, and in particular the time series. There are several aspects of the data that I find hard to understand. Some of them you have addressed, but I am not convinced of the speculations, and some it seems are not discussed much at all. Below a list of open questions:*

*The interpretation of first and second generation compounds: Normally it is of course true that compounds requiring one oxidant attack appear before compounds requiring two. In a situation where you would have a constant oxidant concentration and a constant HOM sink, the interpretation would be as straightforward as presented in this manuscript. However, now neither of these is true. If we focus only on P1, the NO3 concentration drops dramatically when the limonene is added, as does the N2O5, but after that both increase during P1. The oxidation rate of limonene is the source of primary RO2, but this parameter (i.e., limonene\*NO3) is not presented anywhere in the manuscript as far as I can tell. Therefore, I cannot tell what behavior I should expect for the first generation products based on the source strength. The NO3 trace is also plotted on such a scale that it is impossible to read out anything from it. This should be amended, as it is one of the most important parameters in the experiments.*

**Response:**

Accepted. In the revised manuscript, we have revised the scale of NO$_3$ as mentioned above and shown the reaction rate of limonene with NO$_3$ determined using measured limonene and NO$_3$ concentration in Fig. S16e as follows. A measured time series of one typical 1$^{st}$-generation product is also shown. The reaction

rate shows a clear trend similar to 1st-generation reaction products.

[Figure]

●*Similarly, the sinks are always important, and as I stated above, the low wall loss rate seems questionable to me. If indeed the loss rate was so low, I would expect a continuous increase of most products throughout P1, since oxidation is still taking place. I see indication in many figures (e.g. Figs 1, S4, S9) where the drops of the initial peak for some HOM are faster than the wall loss would predict, and this despite additional production still going on, which should cause the drop to be slower than the wall loss rate.*

**Response:**

We think the impression of the reviewer is mainly caused by the too low wall loss rate, given in the manuscript. The time series of a HOM is of course determined by the difference of the source strength (production rate) and sink strength (wall loss rate and further reaction of HOM with $NO_3$). Even if a production is still taking place, "winning" wall loss rates or oxidation rates of HOM will lead to a decrease of the concentration with time rather than a continuous increase.

As mentioned above, the number given for wall loss rate was too low. With updated wall loss rate, the initial drop of the products in Fig. 1, Fig. S4, and Fig. S9 during P1 (the characteristic time of the fastest decay was 15 min, 10 min, and 13 min, respectively) can be explained by the wall loss rate as well as their potential loss by the reaction with $NO_3$ and their sources via the reaction of limonene with $NO_3$.

In the revised manuscript, we have added the following discussion.

"The initial drop of the products in Fig. 1, Fig. S4, and Fig. S9 during P1 (the characteristic time of the fastest decay was 15 min, 10 min, and 13 min, respectively) is attributed to the balance of their sources via the reaction of limonene with $NO_3$, their wall loss, and their potential loss by the reaction with $NO_3$."

●*As an even clearer point relating to the production and loss dynamics, when limonene is added for the second time, many HOM (Figs 1, 4, 5, S3, and in particular S10 and S11) drop very fast, with lifetimes of minutes as far as I can tell. I believe the only time this is addressed is in conjunction with Fig 5 where it is said that this drop coincides with "the onset of particle growth" and thus indicates a role for dimers in NPF. I think the authors need to come up with a convincing line of argumentation why all these HOM seem to drop exactly at that time when limonene is added. The condensation sink does not make a dramatic jump exactly there, and many highly oxygenated monomers and dimers increase again towards the end of P2, when the CS should be still greater, suggesting that CS is not the cause of these changes. In Fig. S9 some dimers even increase dramatically at that time, to later decrease while other types of dimers increase, suggesting that changes in the source strengths are of importance. The dynamics of oxidants and RO2 fates is complex, and I can only guess what is going on, but the data would to me be explainable if the loss rates were much higher than assumed in the manuscript, and the changes mainly governed by changes in source strengths, e.g. total oxidation rate, relative oxidation by O3 vs NO3, and termination by NO3 or RO2.*

**Response:**

The reviewer may have misunderstood the figures, because some details of the figures may not be clear enough. We would like to point out that a few minutes before the second limonene addition, these 2$^{nd}$-generation products had already reached a peak and dropped (Fig. 1, 4, 5, S9, and S10). Therefore, at this time the drop is not related to limonene addition, and thus we explain this with close to zero source of these products at continuously growing condensational sink. We agree with the reviewer that the varying source strength is of importance here. In order to make this clearer, in the revised manuscript, we have enlarged the x-axis of Fig. 1c as follows:

[Figure]

The characteristic time of the fastest decay of the HOM over the 2nd limonene addition in Fig. 1, 4, 5, S9, and S10 is 4-8 min. With the updated wall loss rate, these decays can be explained by the wall loss rate (characteristic time ~8 min) and condensation sink of vapor loss to particles calculated according to the study of Kulmala et al. (2012) (characteristic time ~13 min). The characteristic times of the fastest decay of the HOM at the end of P2 in Fig. S9 and S10 are 1.4-3.4 min, which can also be well explained by the updated wall loss rate and condensation sink of vapor loss to particles at the end of P2 (characteristic time ~1.4 min).

As for later periods (starting from P2), $2^{nd}$-generation products did not stop increasing until the next addition of limonene, indicating that the source leading to these products was strong and had overcome the condensational sink.

In the revised manuscript, we have added the following discussion.

"The characteristic time of the fastest decay of the HOM over the $2^{nd}$ limonene addition in Fig. 1, 4, 5, S9, and S10 are 4-8 min. These decays can be explained by the wall loss rate (characteristic time ~8 min) and condensation sink of vapor loss to particles according to the study of Kulmala et al. (2012) (characteristic time ~13 min). The characteristic times of the fastest decay of the HOM at the end of P2 in Fig. S9 and S10 are 1.4-3.4 min, which can also be well explained by the updated wall loss rate and condensation sink of vapor loss to particles at the end of P2 (characteristic time ~1.4 min)."

●*Concerning RO2 fates, Fig S2 shows loss rates of RO2 radicals at different times. At the same time, 6 ppb limonene was injected but only around 1 ppb is left after 5-10 min, suggesting that 5 ppb limonene (>1e11 cm^-3) has reacted in this 10 min. The loss rates in Fig. S2 suggest lifetimes of around 5 min for RO2 during this period, which means that there should be >1e10 cm^-3 RO2 in the chamber, as each reacted limonene forms an RO2. This concentration seems very high. Or alternatively, the loss rate of RO2 from reactions with other RO2 seems very low. Have the authors considered that the RO2+RO2 reactions may be faster than predicted by the MCM as they are likely to be much more functionalized than the RO2 used to derive the MCM rates? Even for the primary C10H16NO5 RO2, I could expect that the RO2+RO2 reaction were closer to 1e-11 cm^3/s as was found for the primary C10H15O4 RO2 from a-pinene + O3 (DOI: 10.1021/acs.est.8b02210). Much higher than the ~1e-13 cm^3/s given by the MCM for these radicals. How would this impact the interpretations of the manuscript?*

**Response:**

MCM does not include reactions of HOM-RO$_2$•. We agree that for more functionalized RO$_2$•, the RO$_2$• + RO$_2$• reactions could be faster than the rates in MCM. However, the rate coefficients remain unknown, which on top may depend on individual RO$_2$• formula and structure. Currently we do not see a reliable updated set of rate coefficients that are applicable to the reaction system in this study. The rate coefficient of

the self-reaction of $C_{10}H_{15}O_4\bullet$ from α-pinene + $O_3$ is not directly applicable to $C_{10}H_{16}NO_5\bullet$ due to their different functional groups. The concentration of $RO_2\bullet$ reached 1.9 ppb (about $5\times10^{10}$ molecule $cm^{-3}$) after first limonene addition according to our simulation. If the reaction rate constants of $RO_2\bullet + RO_2\bullet$ were higher than those used in MCM, the modeled concentrations of $RO_2\bullet$ would be lower and relative importance of $RO_2\bullet + RO_2\bullet$ would increase.

In the revised manuscript, we have added discussion regarding the influence of the rate of $RO_2\bullet + RO_2\bullet$ on the $RO_2\bullet$ fate as follows:

"We note that the MCM reaction schemes do not include the accretion reactions between $HOM\text{-}RO_2\bullet$. Berndt et al. (2018) determined the rate constant of accretion reaction of $C_{10}H_{15}O_4\bullet$ formed via α-pinene ozonolysis to be $\sim1\times10^{-11}$ $cm^3$ $molecule^{-1}$ $s^{-1}$, which is of the same order as the upper limit for $RO_2\bullet + RO_2\bullet$ reactions used in the MCM schemes for functionalized peroxy radicals such as acyl peroxy radicals (Jenkin et al., 1997; Saunders et al., 2003). However, currently we do not see a reliable updated set of rate coefficients that are applicable to the reaction system in this study. If the rate constants of some $RO_2\bullet + RO_2\bullet$ reactions were higher than those used in MCM, the concentrations of $RO_2\bullet$ would be lower and relative importance of $RO_2\bullet + RO_2\bullet$ in $RO_2\bullet$ fate would increase."

●*Related to many of the points above, like expected behavior for primary and secondary products, the oxidation rates, and the RO2 fates, I suggest that you include some model results in a revised manuscript. For example, it would be very interesting to see how well the MCM run is able to match the measured NO3 and N2O5 behavior, as well as limonene. Likewise, the RO2 concentrations should be included. If the main parameters are captured correctly, I would expect that model could nicely output the time series of NO3\*limonene as well as NO3\*Limonene_oxidation_products, as metrics to show the expected behavior of first and second generation products. If this type of model results match well with the observations (and interpretations), I would be much more convinced.*

**Response:**

In the original manuscript, we had run such an MCM model. In the revised manuscript, we have added a figure in the supplement presenting the results of MCM simulation (Fig. S16):

[Figure]

Figure S16. Simulation results of limonene + NO$_3$ gas-phase chemistry based on MCM v3.3.1 using iChamber model. The whole period of experiment was simulated, at measured T and RH, and additions of limonene, NO$_2$ and O$_3$ were included as initial conditions according to the experimental procedure. (a)~(c): Comparison of simulated (green trace) and measured (red trace) concentrations of (a) NO$_3$, (b) N$_2$O$_5$ and (c) limonene. (d)~(g): Simulated concentrations of (d) total RO$_2$• included in the limonene + NO$_3$ gas-phase chemistry in MCM v3.3.1, (e) rate coefficient k×limonene×NO$_3$, compared with a measured 1$^{st}$-generation product C$_{20}$H$_{31}$NO$_{13}$; (f) "NLIMO2", an example of 1$^{st}$-generation C$_{10}$ RO$_2$•, and (g) "NLIMALO2", an example of 2$^{nd}$-generation C$_{10}$ RO$_2$•, compared with a measured 2$^{nd}$-generation RO$_2$•, C$_{10}$H$_{15}$N$_2$O$_{12}$•.

The modeled concentrations of NO$_3$, N$_2$O$_5$ and limonene by MCM generally match the behavior of measured concentrations (Fig. S16 a~c). The overestimated limonene concentrations can be attributed to the absence of a temperature-dependence of the rate constant for the reaction of limonene with NO$_3$ in MCM. RO$_2$• concentrations showed 1$^{st}$-generation trend (Fig. S16d). The reaction rate (k×limonene×NO$_3$) was highest at every injection of limonene (Fig. S16e). As for oxidation products, the second time of NO$_3$ attack to organic nitrate with a C=C double bond is not included in MCM, so the simulation of the closed-shell products does not present 1$^{st}$ or 2$^{nd}$ generation product patterns as we have observed in CIMS. But we are able to observe several good simulations of 1$^{st}$ and 2$^{nd}$ generation RO$_2$• (Fig. S16f,g). For example, the "NLIMALO2" (Fig. S16g) showed a typical time series of 2$^{nd}$ generation RO$_2$•, which is formed via NO$_3$ attack of a 1$^{st}$-generation carbonyl product which does not contain N atom according to the MCM mechanism.

In the revised manuscript, we have added more results of MCM in the supplement as follows:

"**S3 Simulations based on the Master Chemical Mechanism (MCM)**

Besides simulations of the $RO_2 \cdot$ loss pathway (Sect. 2.5), we conducted several simulations including concentrations of $NO_3$, $N_2O_5$, limonene and $RO_2 \cdot$ (Fig. S16) based on MCM v3.3.1 (http://mcm.york.ac.uk/) using iChamber, an open-source program (https://sites.google.com/view/wangsiyuan/models?authuser=0) (Wang and Pratt, 2017).

The modeled concentration of $NO_3$, $N_2O_5$ and limonene by MCM generally match the behavior of measured concentrations (Fig. S16 a~c). The overestimated limonene concentrations can be attributed to the absence of a temperature-dependent rate constant for the reaction of limonene with $NO_3$. $RO_2 \cdot$ concentrations showed 1st-generation trend ((Fig. S16d). The reaction rate (k×limonene×$NO_3$) was highest at every injection of limonene (Fig. S16e). As for oxidation products, the second time of $NO_3$ attack to organic nitrate with a C=C double bond is not included in MCM, so the simulation of the closed-shell products does not present 1st or 2nd generation product patterns as we have observed in CIMS. But we are able to observe several good simulation of 1st and 2nd generation $RO_2 \cdot$ (Fig. S16f,g). For example, the "NLIMALO2" (Fig. S16g) showed a typical time series of 2nd generation $RO_2 \cdot$, which is formed via $NO_3$ attack of a 1st-generation carbonyl product which does not contain N atom according to the MCM mechanism."

●*The conclusion of the dimers and trimers being important for NPF because they only appear during P1 in Figs 5 and 6 is brought into question when considering that only the N2 dimers (shown in Fig 5) show this feature, while N1, N3 and N4 dimers (Figs S9-S11) all show relatively high concentrations also much later in the experiment, when the CS is far higher than during the transition from P1 to P2. In addition, I am skeptical to the particle concentration trace in Fig. 1b. It keeps increasing up until 2h, but the size distribution gives no indication of new particles being formed after the first hour. Instead, the size distribution seems to barely detect any particles below 20 nm, and the mode just appears at that size. I see no discussion on this point, but the obvious conclusion for me is that the detection limit of the SMPS is about 20 nm, and the particle concentration trace derived from that data is the concentration of particles larger than 20 nm. As such, the particle concentration has likely in reality been decreasing after the initial particle burst during the first minutes or tens of minutes of the experiment. This needs to be discussed.*

**Response:**

We agree that N2 dimers show different patterns from N1, N3 and N4 dimers, which also showed relatively high concentration after P1. In the revised manuscript, we have modified the discussion regarding dimers as follows:

"Generally, other dimers showed similar patterns (Fig. S9-11), though the difference of their concentrations between P2-P6 and P1 were not as large as for the $C_{20}H_{32}N_2O_x$ family. The time when signals of several

dimers (e.g. $C_{20}H_{32}N_2O_x$, $C_{20}H_{33}N_3O_x$, $C_{20}H_{34}N_4O_x$) dropped substantially matched the time of new particle formation (NPF) and the onset of particle growth, indicating that some dimers were likely involved in the early growth of particles."

As for the particle number concentration in Fig. 1b, we realized that only using the data from SMPS is inappropriate. We thank the reviewer for pointing out this limitation. In the revised manuscript, we also show the total number concentration detected by CPC (TSI-3785) which can detect particles over 5 nm. As the reviewer expected, the number concentration detected by CPC reached its peak earlier than that detected by SMPS, about 40 min after the first addition of limonene. For SMPS data, the smallest particle when particles started to appear is at ~15 nm, which can be more clearly seen after we have changed the color bar to log scale in the revised Fig. 1b. Since the size range of our SPMS starts from ~10 nm, this phenomenon is likely attributed to fast growth of new particles from below 10 nm to 15 nm within the time of one SMPS scan (3 min).

In addition, the particle number concentration detected by CPC provides further support of the important role of trimers in NPF. The time when particle number concentrations started to increase matches the time when trimer started to decrease.

The modified Fig. 1b in the revised manuscript is also shown below:

[Figure]

●*Lines 317-319: "it is likely that C10H16NOx radicals converted immediately after their formation". This is a very broad statement, as it includes all radicals, including the primary O5 radical. If this was the case, Fig. S2 would no longer make any sense, since there would be no RO2 around. I would expect very low sensitivity for RO2 radicals with less than 7 O-atoms in the CIMS, and therefore a more likely scenario is perhaps that the primary RO2 either do not efficiently undergo autoxidation, or alternatively, that the RO2 lifetimes were so short that autoxidation was outcompeted.*

**Response:**

By $C_{10}H_{16}NO_x$ radicals, we meant HOM-RO$_2$• radicals. We apologize for this ambiguity. In the revised manuscript, we have changed this to "HOM peroxy radicals $C_{10}H_{16}NO_x$• (x≥6)".

We agree that RO$_2$• radicals as well as other HOM with less than 6 O-atoms have low sensitivity in the NO$_3^-$-CIMS (Riva et al., 2019), which had been stated in our original manuscript. Based on our measurement, we cannot constrain how fast the primary RO$_2$• $C_{10}H_{16}NO_5$• autoxidizes. Regarding the absence of first-generation characteristic of the time profiles of most HOM peroxy radicals $C_{10}H_{16}NO_{x(x>=6)}$•, there are two possible reasons. One is that they did not undergo efficient autoxidation. The other is that they underwent immediate conversion including autoxidation and/or bimolecular reactions such as with other RO$_2$• or NO$_3$ forming closed-shell products or continuing the radical chain to RO•. The instantaneous increase of 2N-dimers and trimers after the first limonene addition suggests that $C_{10}H_{16}NO_{x(x>=6)}$• were indeed formed efficiently via autoxidation. Therefore, we think that the latter reason is more likely. We have revised the original sentence as follows:

> *"The absence of first-generation characteristics of the time profile of most HOM peroxy radicals $C_{10}H_{16}NO_{x(x>=6)}$• may be attributed to two possible reasons. They either did not undergo efficient autoxidation, or they underwent immediate conversion including autoxidation and/or bimolecular reactions with other RO$_2$• or NO$_3$ forming closed-shell products such as dimers or continuing the radical chain forming RO•. The instantaneous increase of 2N-dimers and trimers after the first limonene addition shown below suggests that $C_{10}H_{16}NO_{x(x>=6)}$• were indeed formed efficiently via autoxidation. Therefore, the latter reason is more likely."*

●*Lines 452-453: "the most abundant C20H32N2Ox was expected to have an oxygen number of 18 according to the accretion reaction mechanism". At least I would not "expect" this, as seemingly all earlier studies discussing this topic have suggested that the dimers have lower O/C because the less oxidized RO2 radicals are involved. As you also cite 4 papers for this, the "expected" should be removed.*

**Response:**

Accepted. We meant the "expected" formula if all dimer were formed by accretion reactions of only HOM RO$_2$•. In the revised manuscript, we have revised this sentence as follows:

> "…the most abundant $C_{20}H_{32}N_2O_x$ would have an oxygen number of 18 if they were exclusively formed by the accretion reaction of HOM RO$_2$•."

We have also updated the references here to include only the papers that explicitly address whether less oxidized RO$_2$• radicals are involved in accretion reactions.

●*Lines 570-571: "in the early stage of the experiment when new particle formation (NPF) had not occurred yet". As I said earlier, the time when NPF occurred is not visible from your data, and likely happened much earlier than the SMPS concentrations indicates. In addition, you have said that trimers are the most likely candidates to initiate nucleation, and the trimers formed almost instantly, suggesting that NPF would also start immediately.*

**Response:**

Accepted. In the revised manuscript, we have changed this sentence to:

"Dimers contributed 47 % in the early stage of the experiment when  particle surface concentration was rather low ($< 6\times10^4$ nm$^2$ cm$^{-3}$), which was similar to monomers (47 %)."

●*Line 605: How do you come to these numbers? At 1 ppb limonene, the lifetime should be <5s.*

**Response:**

This is a typo. We meant $RO_2\bullet$ instead of $NO_3$ and we have corrected this in the revised manuscript.

*Technical corrections*

●*The term "SOA growth" is used several times in the abstract, and it is confusing to me, as would be terms like "sulfate growth" or "black carbon growth".*

**Response:**

The acronym SOA contains the word aerosol. That is different from the two examples given by the reviewer. Nevertheless, we have changed all the terms "SOA growth" to "growth of SOA particles".

●*Lines 179-180: I suggest not to mention this 14% uncertainty here, since it gives the picture that this is the total uncertainty, although that is given later.*

**Response:**

We add the word "additional" to clarify this number:

"A mass-independent transmission efficiency was used according to our previous study, which causes an additional uncertainty of 14 %."

●*Use of %: In the abstract the Hom yield is given as "2.5 % (+1.7 %/-0.7 %)". It is not trivial to understand this. I assume that it means 0.8-3.2 % as the uncertainty range, but more commonly +- X% would mean percentage and not percentage points. On line 183, are these percentages or percentage points?*

**Response:**

We meant percentage points. In the revised manuscript, we have changed the form to superscripts and subscripts to avoid ambiguity:

"$1.5\ \%^{+1.7\ \%}_{-0.7\ \%}$"

●*Figure 3: An O-atom based Kendrick plot is unlikely to be obvious to the majority of readers. Please add somewhere a sentence about what the y-axis means.*

**Response:**

The explanation of O-atom based Kendrick mass defect has been added in the annotation of Figure 3 in the revised manuscript as follows.

"The calculation of O-atom-based Kendrick mass defect includes two steps. First, the IUPAC mass scale (based on the $^{12}C$ atomic mass as exactly 12 Da) is rescaled to Kendrick mass: Kendrick mass = IUPAC mass × (16/15.9949), which converts the mass of O from 15.9949 to exactly 16. Then, Kendrick mass defect is given by: Kendrick mass defect = nominal Kendrick mass – exact Kendrick mass. Thus, compounds with the same number of each kind of atom except for O have equal O-atom-based Kendrick mass defect, and are shown in a horizontal line in the O-atom-based Kendrick mass defect plot."

●*Lines 358-360: What about the carbonyl formation from alkoxy radicals?*

**Response:**

We have added this pathway and discussed it as follows:

"As discussed in our previous study by Shen et al. (2021), this higher abundance of carbonylnitrates is not likely to be explained by the reaction of alkoxy RO• + $O_2$ forming carbonyls and $HO_2$•, decomposition of β-nitrooxyperoxynitrate or self-reactions of $RO_2$• via the Bennett and Summers mechanism forming carbonyls and $H_2O_2$. Reactions between $RO_2$• in general should produce overall equal amounts of carbonyl and hydroxyl compounds. The decomposition of β-nitrooxyperoxynitrate is slow in the gas-phase. The reaction of alkoxy RO• with $O_2$ for large RO• is generally slower than isomerization and decomposition (Vereecken

and Peeters, 2009, 2010). Thus, the higher abundance of carbonylnitrates compared to hydroxynitrates may be attributed to unimolecular termination of HOM-RO$_2\bullet$. In addition, isomerization of RO$\bullet$ forming carbonyl compounds may also contribute to this finding."

●*Figure 5: Why is the peak intensity on the right axis in the top panel? It seems strange to me, and it took me a long while to realize what the arrows in the plot meant.*

**Response:**

In the revised manuscript, we have moved the peak intensity to the left axis.

●*Lines 447-448: While correct, this sentence seems to suggest a causality opposite to what one would expect.*

**Response:**

We wrote this sentence simply to state our observation based on the time profile of $C_{20}H_{32}N_2O_x$ rather than suggest a causality. In the revised manuscript, we have revised this sentence as follows:

"The relative contribution of second-generation formation was observed to increase with increasing oxygen number."

●*Line 582: I think "N" is missing.*

**Response:**

We thank the reviewer for pointing out this typo. It should be $C_{10}H_{16}NO_5\bullet$.

●*Line 609: "Volatile commercial products"?*

**Response:**

VCP is the abbreviation for volatile chemical products, which can be referred to work of Mcdonald et al. (2018).

**References**

[revised manuscript text omitted]

---

## Author Comment (AC2)

We thank the reviewer for the comments on our manuscript. The comments and suggestions are greatly appreciated. All the comments have been addressed and we believe that the revisions based on these comments improved the quality of our manuscript. Below please find our responses to the comments one by one and the corresponding revisions made to the manuscript. The original comments are shown in italics. The revised parts of the manuscript are highlighted.

*This study provided a detailed analysis of the HOM formation from limonene reacting with NO3 radical. These HOMs are potentially important in forming new particles and secondary organic aerosol. In addition, this study provides observational evidence on the formation of HOM-trimer from NO3 oxidation of limonene, to my best knowledge, for the 1st time. However, I have a few comments to be addressed before this manuscript can be published in ACP.*

1. *The concentration issue: The monoterpene concentrations are still much higher than in most regions. This could be why the dimer/monomer ratio is so high, and why can you observe trimer? However, this is far from the real atmosphere, where dimers concentration is usually too low to react with the other oxidant before condensing to particles. As shown in Fig.5 and Fig. 6, some aerosol exists universally in the atmosphere; they can scavenge HOM dimers so effectively that the dimers have no time to react with oxidants again to form a trimer.*

**Response:**

We agree that the higher monoterpene concentrations may favor trimer formation. Although monoterpene concentrations in this study (0-0.92 ppbv) are higher than in most ambient regions, they are still in the range of ambient concentrations (~0.01-1 ppbv) (e.g. Coggon et al., 2021; Wang et al., 2022), especially for forested regions (e.g. Xu et al., 2015; Kontkanen et al., 2016; Janson, 1992).

Regarding the influence of aerosols, the relative loss of dimers by condensations on aerosols and by gas-phase reaction depends on the concentrations of aerosol and oxidants. Assuming that dimers react with $NO_3$ at a rate constant similar to limonene and have a condensation sink similar to $H_2SO_4$ ($10^{-3}$ to $10^{-1}$ s$^{-1}$) (Dada et al., 2020) at a $NO_3$ concentration of 5-300 ppt, the lifetime with respect to $NO_3$ and to condensation on particles are ~0.1-10 min and ~0.1-20 min, respectively. Therefore, although aerosols may scavenge HOM dimers hindering the formation of trimers, dimers can still react with oxidant, at least with $NO_3$ at nighttime, forming trimers. Such reactions are particularly important when the ambient aerosol concentration is low (similar to the situation of the early stage of our experiment).

In the revised manuscript, we have added the following discussion regarding the VOC concentrations:

"Since our experiment of $NO_3$ oxidation of limonene was performed under near atmospheric conditions, such NPF events induced by the oxidation of limonene by $NO_3$ could also occur in the ambient atmosphere. Although monoterpene concentrations in this study (0-0.92 ppbv) are higher than in most ambient regions, they are still in the range of ambient concentrations (~0.01-1 ppbv) (e.g. Coggon et al., 2021; Wang et al., 2022), especially for forested regions (e.g. Xu et al., 2015; Kontkanen et al., 2016; Janson, 1992). Assuming that dimers react with $NO_3$ at a rate similar to limonene, and that they have a condensation sink similar to $H_2SO_4$ ($10^{-3}$ to $10^{-1}$ $s^{-1}$) (Dada et al., 2020), the lifetime with respect to $NO_3$ at an $NO_3$ concentration of 5-300 ppt and to condensation on particles are ~0.1-10 min and ~0.1-20 min, respectively. Therefore, although aerosols may scavenge HOM dimers in the ambient atmosphere, dimers can still react with $NO_3$ at nighttime, forming trimers. Such reactions are particularly important when the ambient aerosol concentration is low."

2. *Self-termination of HOM-RO2: One of this work's major conclusions is that the HOM-RO2 self-termination is more important than the previously understood. I believe more evidence is needed to support this point. For example, is it possible C10H15NOx is formed from C10H15NOx+NO3? In addition, what's the potential influence of the differences in instrument sensitivity on detecting carbonyl compounds and hydroxyl compounds?*

**Response:**

We observed much more carbonylnitrates than hydroxynitrates (or hydroperoxynitrates) in the gas-phase products in the limonene + $NO_3$ reaction system. As discussed in our previous study by Shen et al. (2021), this higher abundance of carbonylnitrates is not likely to be explained by the reaction of alkoxy RO• + $O_2$ forming carbonyls and $HO_2$•, decomposition of β-nitrooxyperoxynitrate or self-reactions of $RO_2$• via the Bennett and Summers mechanism forming carbonyls and $H_2O_2$. Reactions between $RO_2$• in general produce overall equal amounts of carbonyl and hydroxyl compounds. The decomposition of β-nitrooxyperoxynitrate is slow in the gas-phase. The reaction of alkoxy RO• with $O_2$ for large RO• is generally slower than isomerization and decomposition. Thus, the higher abundance of carbonylnitrates than hydroxynitrates may be attributed to unimolecular termination of HOM-$RO_2$•. In addition, isomerization of RO• forming carbonyl compounds may also contribute to this finding.

A $C_{10}H_{15}NO_x$ compound could be formed via the reaction of $C_{10}H_{15}NO_{x-2}$ with $NO_3$ and $O_2$ followed by the conversion of $RO_2$• to RO• and the dissociation forming $NO_2$. However, this reaction does not have an impact on the total amount of $C_{10}H_{15}NO_x$, which still need a source. Also, main products of $C_{10}H_{15}NO_x+NO_3$ are compounds containing 2 N atoms.

According to Hyttinen et al. (2015), for nitrate CI-APi-TOF, HOM containing two hydrogen bond donors

(such as -OOH and -OH group) have strong binding energy with $NO_3^-$. Additional hydrogen bond donors only enhance the binding energy marginally. If we compare HOM carbonyl product (such as $C_{10}H_{15}NO_{10}$) with the corresponding hydroxy product ($C_{10}H_{17}NO_{10}$), they only differ in one functional group. As both are highly functionalized, it is likely that carbonyl HOM have a quite similar sensitivity with hydroxyl HOM. If the sensitivity of carbonyl HOM were lower, this would result in even more dominance of carbonyl HOM over hydroxyl HOM. Thus, we conclude that carbonylnitrates are more abundant than hydroxynitrates or hydroperoxynitrates.

In the revised manuscript, we have added discussion on this point as follows.

"As discussed in our previous study by Shen et al. (2021), this higher abundance of carbonylnitrates is not likely to be explained by the reaction of alkoxy RO• + $O_2$ forming carbonyls and $HO_2$•, decomposition of β-nitrooxyperoxynitrate or self-reactions of $RO_2$• via the Bennett and Summers mechanism forming carbonyls and $H_2O_2$. Reactions between $RO_2$• in general should produce overall equal amounts of carbonyl and hydroxyl compounds. The decomposition of β-nitrooxyperoxynitrate is slow in the gas-phase. The reaction of alkoxy RO• with $O_2$ for large RO• is generally slower than isomerization and decomposition (Vereecken and Peeters, 2009, 2010). Thus, the higher abundance of carbonylnitrates compared to hydroxynitrates may be attributed to unimolecular termination of HOM-$RO_2$•. In addition, isomerization of RO• forming carbonyl compounds may also contribute to this finding."

"According to Hyttinen et al. (2015), for nitrate CI-APi-TOF, HOM containing two hydrogen bond donors (such as -OOH and -OH group) have strong binding energy with $NO_3^-$. Additional hydrogen bond donors only enhance the binding energy marginally. If we compare HOM carbonyl product (such as $C_{10}H_{15}NO_{10}$) with the corresponding hydroxy product ($C_{10}H_{17}NO_{10}$), they only differ in one functional group. As both are highly functionalized, it is likely that HOM carbonyl have a quite similar sensitivity with hydroxyl HOM. If the sensitivity of carbonyl HOM were lower, this would result in even more dominance of carbonyl HOM over hydroxyl HOM. Thus, we conclude that carbonylnitrates are more abundant than hydroxynitrates or hydroperoxynitrates."

3. *I would suggest adding more discussions on the potential influence of ozone oxidation of limonene in the system, as well as the potential role of HO forming via ozonolysis of monoterpene. As shown in Table 2, C20H33NOx and C20H34N4Ox are likely from OH oxidation. In addition, the abundance of C20H31NOx (x=10-15) is considerably high may also indicate the role of O3 chemistry.*

**Response:**

Accepted. We agree that $C_{20}H_{33}NO_x$ may be formed via accretion reaction of $C_{10}H_{16}NO_x$• with $C_{10}H_{17}O_x$•, which is formed via OH oxidation of $C_{10}H_{16}$. $C_{20}H_{34}N_4O_x$ may be formed via accretion reaction of two

$C_{10}H_{17}N_2O_x\bullet$, which can be formed via OH oxidation of $C_{10}H_{16}N_2O_x$. Also, the considerably high abundance of $C_{20}H_{31}NO_x$ (x=10-15) may be partly attributed to the contribution of $O_3$ chemistry. The revised texts are as follows:

"We cannot exclude that the formation pathway of $C_{20}H_{33}NO_x$, $C_{20}H_{34}N_4O_x$ and $C_{19}H_{31}NO_x$ may also involve limonene oxidation by OH• (Table 2), which can be formed in the ozonolysis of limonene as a minor pathway. In addition, the high abundance of $C_{20}H_{31}NO_x$ (x=10-15) among the dimers may be partly attributed to a contribution of the reaction of limonene with $O_3$."

We have also updated Table 2 to reflect the potential contribution of OH oxidation as follows:

Table 2. Major dimer and trimer families and their possible formation pathways.

| Dimer/Trimer family | Possible formation pathways |
|---|---|
| $C_{20}H_{32}N_2O_x$ | $C_{10}H_{16}NO_x\bullet + C_{10}H_{16}NO_x\bullet$ |
| $C_{20}H_{33}N_3O_x$ / $C_{20}H_{31}N_3O_x$ | $C_{20}H_{32}N_2O_x + NO_3 + HO_2\bullet/RO_2\bullet$ |
| $C_{20}H_{31}NO_x$ | $C_{10}H_{16}NO_x\bullet + C_{10}H_{15}O_x\bullet$ |
| $C_{20}H_{33}NO_x$ | $C_{10}H_{16} + OH\bullet + C_{10}H_{16}NO_x\bullet$ |
| $C_{20}H_{34}N_4O_x$ | $(C_{10}H_{16}N_2O_x + OH\bullet) + (C_{10}H_{16}N_2O_x + OH\bullet)$ |
| $C_{19}H_{30}N_2O_x$ | $C_{10}H_{16}NO_x\bullet + C_9H_{14}NO_x\bullet$ |
| $C_{19}H_{31}N_3O_x$ | $C_{19}H_{30}N_2O_x + NO_3 + HO_2\bullet/RO_2\bullet$ |
| $C_{19}H_{29}NO_x$ | $C_9H_{14}NO_x\bullet + C_{10}H_{15}O_x\bullet$ |
| $C_{19}H_{31}NO_x$ | $C_{10}H_{16} + OH\bullet + C_9H_{14}NO_x\bullet$ |
| $C_{30}H_{48}N_4O_x$ | $C_{20}H_{32}N_3O_x\bullet + C_{10}H_{16}NO_x\bullet$ |
| $C_{30}H_{47}N_3O_x$ | $C_{20}H_{31}N_2O_x\bullet + C_{10}H_{16}NO_x\bullet$ |

4. *There are so many details in the study, which is good, but makes the manuscript not so easy to follow. I suggest adding some summary statement in each section.*

**Response:**

Accepted. We have added a summary statement at the end of each sub-section in the result section.

*Detailed comments:*

5. *There is growing evidence that monoterpene-OOMs are also important in urban regions. I suggest adding some discussion in the introduction part, i.e., Liu et al., 2021, ACP; Nie et al., 2022, Nat. Geosci.*

**Response:**

Accepted. The revised texts are as follows:

"Besides the observations at forested regions, monoterpene derived HOM via $NO_3$ oxidation also contribute to organic aerosols in urban regions. For example, Liu et al. (2021) and Nie et al. (2022) have found that HOM derived from monoterpene nighttime chemistry are important in megacities in China, especially during summertime."

6. *Line 162: what's the concentration of isoprene-HOMs in the chamber? Can they influence the subsequent reactions?*

**Response:**

The concentration of isoprene-HOM in our chamber are less than 1 ppt. Besides, all the isoprene-HOM observed ($C_5H_9NO_{7,10}$, $C_5H_8N_2O_{8-10}$, $C_5H_{10}N_2O_8$, $C_5H_9N_3O_{9,10}$) are saturated and do not contain C=C double bond. Therefore, the isoprene-HOM will not influence the reaction of limonene with $NO_3$ in this study.

In the revised manuscript, we have added the following discussion:

"These isoprene oxidation products were likely formed in an isoprene + $NO_3$ experiment performed two days before (Zhao et al., 2021) and released slowly from chamber walls due to their semi-volatile character. Their total concentration is less than 1 ppt. All the isoprene-HOM observed ($C_5H_9NO_{7,10}$, $C_5H_8N_2O_{8-10}$, $C_5H_{10}N_2O_8$, $C_5H_9N_3O_{9,10}$) are saturated and do not contain C=C double bond. The isoprene-HOM will not influence the reaction of limonene with $NO_3$ in this study. Therefore, they are not discussed as products from the limonene oxidation in our experiment."

7. *Line 179-180: More discussion on the mass-independent transmission calibration rather than citing a reference.*

*Calibration issue: the authors quantify observed HOMs by calibrating H2SO4 and assuming they have similar charging efficiency. However, besides charging efficiency, the transmission would influence the calibration coefficient either (Junninen et al., 2010), especially when referring to a bunch of the molecules covering a wide mass range. Like in this study, one can reasonably speculate the transmission between HOM-dimers and HOM-monomers is different. I may recommend reconsidering the calibration factors used in the current version.*

**Response:**

Accepted. We have added discussion about the dependence of transmission on m/z as follows:

"A mass-independent transmission efficiency was used according to our previous study, which causes an additional uncertainty of 14 % (Pullinen et al., 2020). In this previous study, the transmission efficiency curve of nitrate CI-APi-TOF was determined and found to monotonously decrease with increasing mass of ions but only slightly depend on the mass range (14% change). As we used the same setting as our previous study, we have included the slight dependence of transmission on m/z in the uncertainties."

8. *How to calculate the wall loss of N2O5 of the chamber?*

**Response:**

The wall loss rate constant of $N_2O_5$ in the SAPHIR chamber is $7.2\times10^{-5}\,s^{-1}$ (Fry et al., 2009), and the lifetime of $N_2O_5$ due to wall loss is about 4 h. As the HOM yield determination is based on the first 3 min, the wall loss of $N_2O_5$ can be ignored compared with the loss via the reaction of $NO_3$ with limonene.

In the revised manuscript, we have added the following note.

"The wall loss rate constant of $N_2O_5$ in the SAPHIR chamber is $7.2\times10^{-5}\,s^{-1}$ (Fry et al., 2009). As the HOM yield determination is based on the first 3 min, the wall loss of $N_2O_5$ can be ignored compared to the loss via the reaction of $NO_3$ with limonene."

9. *Line 209-212: SVOCs can also contribute to SOA formation, especially in the case when SOA monotonic increases.*

**Response:**

We agree that SVOC can contribute to SOA formation, especially when particle concentrations are relatively high. However, according to our analysis (Sect. 3.6), the LVOC and ELVOC already constitute most of the SOA in this study and the fraction of SVOC in total SOA must be low. Therefore, the contribution of SVOCs to SOA formation is likely small in this study, especially during the early stage of particle formation.

In the revised manuscript, we have modified this sentence as follows.

"In Scenario 2, only the irreversible uptake of LVOC and ULVOC/ELVOC compounds were considered to contribute to the growth of SOA particles in order to examine the role of LVOC and ELVOC while IVOC and SVOC were not included, although they may also contribute to SOA."

10. *Please mark clearly of P1 to P6 in Figure 1. The current version makes the statements in the text a bit hard to follow.*

**Response:**

Accepted. In the revised manuscript, Fig. 1a has been modified in which P1 to P6 as well as P1a are marked clearly (also shown below).

[Figure]

11. *Line 278-279: why there was only one peak of C10H15NO9?*

**Response:**

As we pointed out in the text, we could not explain the behavior of $C_{10}H_{15}NO_9$ in section 3.2.2: "At this time, we do not have a reasonable explanation for the trend of $C_{10}H_{15}NO_9$, though we should consider that there are many isomers at play, which may have very different chemical pathways (un)available." We suspect that $C_{10}H_{15}NO_9$ may contain multiple isomers, which follow patterns of different generation products and together result in the general trend of only one peak with time. However, we cannot justify this speculation.

12. *Line 296: why the pattern is 16 TH intervals other than 32 TH intervals?*

**Response:**

According to autoxidation mechanism, the pattern should be 32 Th intervals. However, $RO_2\bullet$ can be transformed to $RO\bullet$ via reactions with NO, other $RO_2\bullet$, or $NO_3$, which leads to a pattern of 16 Th interval.

In the revised manuscript, we have further clarified this point as follows.

"Such a pattern is attributed to autoxidation of $RO_2\bullet$ (with 32 Th interval for each $O_2$ addition) plus the alkoxy-peroxy pathway (shifted by 16 Th compared with exclusive autoxidation) as discussed below."

13. *Line 410: Can C10H14Ox be formed from proposed NO3 oxidation pathways?*

**Response:**

According to scheme S2, we can only assume that $C_{10}H_{16}O_x$ may be formed via $NO_3$ oxidation of limonene. We are not aware of pathways to form $C_{10}H_{14}O_x$ via $NO_3$ oxidation of limonene, to our knowledge.

*14. Add ULVOC in Fig. 7, and explain why dimer cannot trigger NPF?*

**Response:**

Accepted. We have added ULVOC in Fig. 7b (also shown below).

[Figure]

In this study, NPF was observed as well as HOM trimers. In contrast, in a previous study investigating the $NO_3$-initiated oxidation of β-pinene also conducted in the SAPHIR chamber under similar conditions, barely NPF was observed (Shen et al., 2021), as we discussed in our original manuscript. Also in that study, no trimers were observed. Therefore, NPF in our study was more likely attributed to HOM trimers since they have the strongest potential of initiating nucleation due to their much lower volatility compared to dimers. We did not intend to state that dimers cannot trigger nucleation. Under the same conditions, trimers are more likely to trigger NPF than dimers. In the revised manuscript, we have further clarified this point as follows. "Therefore, NPF in the current study can more likely be attributed to HOM trimers since they have the strongest potential of initiating nucleation, although we cannot rule out some contribution of dimers in the NPF."

**References**

[revised manuscript text omitted]

---

## Author Response (AR2)

We thank the reviewer for the helpful comment on our manuscript. Below please find our response to the comment and the corresponding revisions made to the manuscript. The original comment is shown in italics. The revised parts of the manuscript are highlighted.

*The authors have nicely addressed the vast majority of my concerns, and I recommend publication. The corrected (higher) wall loss rate makes much more sense, and helps to explain many of the results.*

*My only remaining point relates to the response highlighted in yellow on page 12 of the responses. While the high loss rates leading to the fast (few min) decays are clear, as the authors describe, the actual time series is still not obviously described based on this alone. The high loss rate should have been almost equally high just before the limonene injection as well, yet the signals were still high. The fact that we see a large decay when limonene is added means that the loss rate is high AND that the source is suddenly turned off. Thus, the more interesting question to me is why the source of these HOM is turned off at the point when limonene+NO3 reactions increase dramatically. I do not think that the authors answer this question in the current manuscript. And I think some interesting understanding of the reaction dynamics might lie behind this answer. However, I leave it to the consideration of the authors how far they wish to delve into this issue. At the very least, this effect of the sudden decay should be commented from the perspective of the HOm source as well.*

**Response:**

Accepted. We thank the reviewer for pointing out this problem. We agree that the rapid decays of some HOM at the time of limonene injection indicate the sudden decreases of the source. As these HOM are mostly 2$^{nd}$-generation products, an explanation could be that $NO_3$ is rapidly consumed at limonene injection, suppressing the formation of 2$^{nd}$-generation products.

We have further discussed this problem in page 21 of the revised manuscript as follows:

"…… The characteristic times of the fastest decay of the HOM at the end of P2 in Fig. S12 and S13 are 1.4-3.4 min, which can also be well explained by the updated wall loss rate and condensation sink of vapor loss to particles at the end of P2 (characteristic time ~1.4 min). ==In addition, for some HOM (e.g., $C_{10}H_{15}NO_{10}$ (Fig. 1), $C_{10}H_{14}N_2O_{10}$ (Fig. 4), $C_{20}H_{33}N_3O_{17}$ (Fig. S13), $C_{20}H_{34}N_4O_{17}$ (Fig. S14)), the times of limonene additions (except for the first time) matched the time when HOM signals dropped rapidly. This phenomenon implies a sudden decrease of the source of these HOM at limonene additions as sinks including the losses to walls and particles were largely invariant in such a short time. The decrease of the source may be attributed to the rapid depletion of $NO_3$ at limonene injections (Fig. 1). As many of these HOM are 2$^{nd}$-generation products, i.e. formed via the reactions of 1$^{st}$-generation products with $NO_3$, the depletion of $NO_3$ could lead to sudden==

decreases of the source of the 2nd-generation HOM, which accounted for most HOM in the study."